# Enabling nanoscale flexoelectricity at extreme temperature by tuning cation diffusion

Leopoldo Molina-Luna [1], Shuai Wang [2], Yevheniy Pivak[3], Alexander Zintler[1], Héctor H. Pérez-Garza[3], Ronald G. Spruit [3], Qiang Xu[3,4], Min Yi [2], Bai-Xiang Xu [2] & Matias Acosta [5]

Any dielectric material under a strain gradient presents flexoelectricity. Here, we synthesized 0.75 sodium bismuth titanate −0.25 strontium titanate (NBT-25ST) core–shell nanoparticles via a solid-state chemical reaction directly inside a transmission electron microscope (TEM) and observed domain-like nanoregions (DLNRs) up to an extreme temperature of 800 °C. We attribute this abnormal phenomenon to a chemically induced lattice strain gradient present in the core–shell nanoparticle. The strain gradient was generated by controlling the diffusion of strontium cations. By combining electrical biasing and temperature-dependent in situ TEM with phase field simulations, we analyzed the resulting strain gradient and local polarization distribution within a single nanoparticle. The analysis confirms that a local symmetry breaking, occurring due to a strain gradient (i.e. flexoelectricity), accounts for switchable polarization beyond the conventional temperature range of existing polar materials. We demonstrate that polar nanomaterials can be obtained through flexoelectricity at extreme temperature by tuning the cation diffusion.

[1] Department of Materials and Earth Sciences, Advanced Electron Microscopy (AEM) Group, Technische Universität Darmstadt, Alarich-Weiss-Strasse 2, 64287 Darmstadt, Germany. [2] Department of Materials and Earth Sciences, Mechanics of Functional Materials Division, Technische Universität Darmstadt, Otto-Berndt-Strasse 3, 64287 Darmstadt, Germany. [3] DENSsolutions, Informaticalaan 12, 2628ZD Delft, Netherlands. [4] Kavli Centre of NanoScience, National Centre for HRTEM, TU Delft, 2628CJ Delft, Netherlands. [5] Department of Materials and Earth Sciences, FG Nichtmetallische-Anorganische Werkstoffe, Technische Universität Darmstadt, Alarich-Weiss-Strasse 2, 64287 Darmstadt, Germany. These authors contributed equally: Leopoldo Molina-Luna, Shuai Wang. Correspondence and requests for materials should be addressed to L.M.-L. (email: molina@geo.tu-darmstadt.de) or to M.A. (email: ma771@cam.ac.uk)

Materials with switchable polarization are indispensable in memory devices[1], sensors[2], actuators[3], and transducers[4]. Polarization in dielectrics can be induced by different stimuli, e.g., mechanical strain via piezoelectricity. In contrast to piezoelectricity, which requires a non-centrosymmetric crystal structure, there is an intrinsic property in any dielectric material that can generate polarization under a strain gradient. It is referred to as flexoelectricity[5,6].

Flexoelectricity has been first studied in liquid crystals[7] and has recently gained widespread interest for a broad range of material classes like ferroelectrics[8,9], semiconductors[10] and biomaterials[11]. Many electromechanical[8,12,13] and memory[9] devices have been realized using the flexoelectric effect. Theoretical models show that the flexoelectric response scales inversely with size[14] and thus it plays a significant role especially at the nanoscale[14–16]. Designing flexoelectric nanomaterials without the need for an external mechanical load has been previously reported in compositionally graded materials[17,18] and domain wall engineered ferroelectric thin films[8].

Previous studies[19–21] showed that compositional gradients can be obtained by utilizing the diffusion of strontium cations in bulk ceramics and nanoparticles of 0.75 sodium bismuth titanate −0.25 strontium titanate (NBT-25ST). By making use of this compositional gradient, a strain inhomogeneity and therefore, a flexoelectric-based polarization can be induced. In order to directly observe the polarization and polarization switching under an applied electric field structural data needs to be acquired with high spatial resolution. In situ transmission electron microscopy (TEM) studies under simultaneous electric and temperature stimuli would be the method of choice.

In this communication, we report an abnormal phenomenon, the presence of domain-like nanoregions (DLNRs) in a NBT-25ST nanoparticle at extreme temperature. These DLNRs are stable above the Burns temperature and change with an applied electric field. Several mechanisms, e.g. piezoelectricity electrostriction and ferroelectricity, have been excluded as main factors for the observed physical phenomenon. Instead, we ascribe the origin of the DLNRs to flexoelectricity. The strain gradient in the nanoparticles was generated by controlling the slow lattice diffusion of strontium cations. Our claim is supported by comparing the in situ TEM results with phase field simulations. The phase field simulations yield a similar polarization distribution when the flexoelectric effect is considered. Our results provide a novel way to generate flexoelectric-induced polarization and a simple yet effective route to design polar nanomaterials with a built-in strain gradient using cation diffusion.

## Results

**In situ heating and electrical biasing.** In order to observe and manipulate a strain gradient at the nanoscale, we directly synthesized core–shell NBT-25ST nanoparticles inside a TEM. The solid-state solution NBT-25ST was chosen as the model system for three reasons. First, NBT-25ST has a high dielectric constant, which gives rise to a high flexocoupling coefficient. Second, the $Sr^{2+}$ diffusion can be accelerated or suppressed dramatically by modifying the A-site stoichiometry[21]. This renders an attractive system to tune cation diffusion and to generate a strain gradient without additional mechanical loading. Last, the nanoscale (~100 nm) synthesis of particles makes it easier to generate large strain gradients[16].

We used a microelectromechanical (MEMS) based electrothermal nano-chip for the in situ TEM experiments[22]. The nano-chip consists of an encapsulated microheater and electrical biasing electrodes (Fig. 1a, c). A suspension of raw powders with a stoichiometry of 75 mol% $Na_{1/2}Bi_{1/2}TiO_3$-25 mol% $SrTiO_3$ was placed on a silicon nitride electron transparent window. The powders were heated using a defined temperature profile ramp based on a previous study[19] (described in Supplementary Figure 1). With this approach, we synthesized core–shell NBT-25ST nanoparticles directly inside the TEM. In order to estimate the electric field and temperature distribution generated between the electrical biasing electrodes, we performed a finite element analysis using COMSOL Multiphysics. Figure 1b, d shows that the temperature and the applied electric field between the electrodes are homogeneous at the sample region. By using this

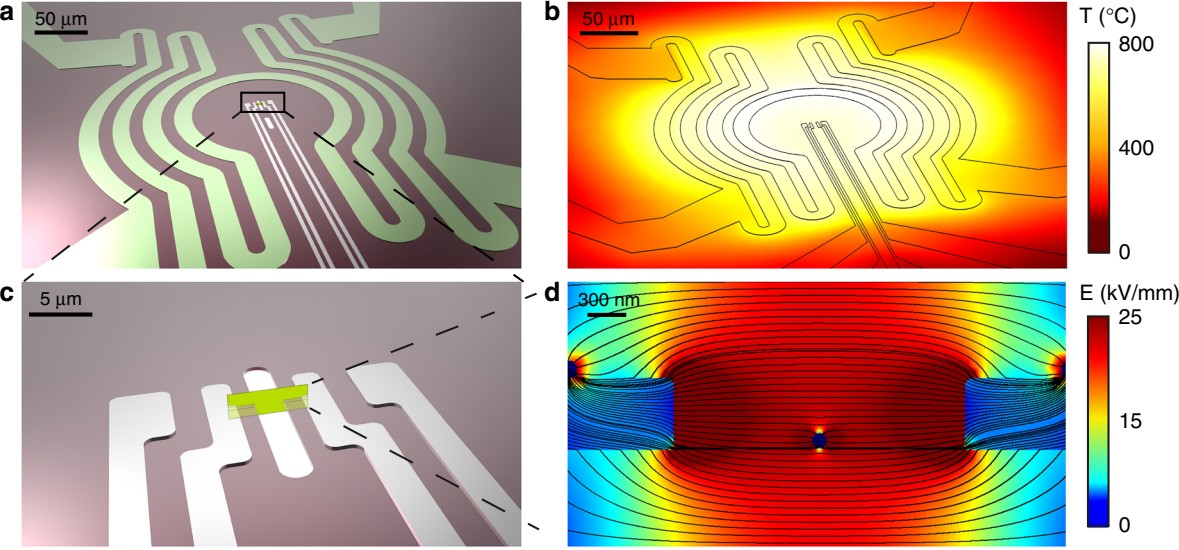

**Fig. 1** Electro-thermal chip sample carrier for in situ transmission electron microscopy. **a** Schematics of the electro-thermal chip, including the set of biasing electrodes surrounded by the encapsulated microheater colored in green that is temperature controlled by Joule heating. **b** Corresponding simulated temperature distribution profile generated by the microheater. **c** Magnified view of the biasing wires region, showing a close-up of the 20 nm thick electron transparent window and the four surrounding biasing wires. The green plane represents the cross-section where the electric field magnitude is plotted. **d** Finite element simulation of local electric field magnitude and the electric field lines over the cross-sectional plane indicated in **c**. A nanoparticle was placed in the window area between the electrodes for modeling

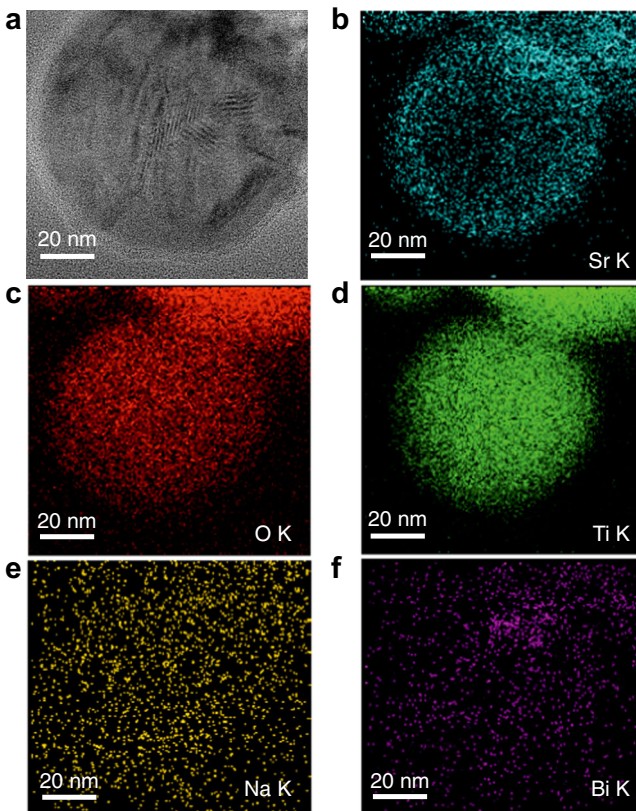

**Fig. 2** Compositional elemental mapping of a core–shell nanoparticle. **a** Bright-field transmission electron microscopy (TEM) image of a core–shell nanoparticle where some domain-like nanoregions (DLNRs) can be observed. **b** Scanning TEM energy-dispersive X-ray spectroscopy elemental mapping of the strontium-K ionization edge. It indicates there is a strontium enrichment in the shell and a strontium deficiency in the core. **c**–**f** Elemental maps of the remaining homogeneously distributed elements

experimental setup, it was possible to observe physical processes with atomic resolution while applying a temperature of up to 800 °C and simultaneously setting a potential of up to 100 V. This bias corresponds to an electric field of up to ~20 kV mm$^{-1}$ for electrodes separated by a 5 µm gap.

By using scanning transmission electron microscopy (STEM) in combination with energy-dispersive X-ray spectroscopy (EDS), we could monitor the core–shell NBT-25ST formation process in situ at $T = 300$, 600, and 800 °C. After performing a careful structural analysis at 800 °C on several nanoparticles we observed the formation of DLNRs. An exemplary single nanoparticle is shown in Fig. 2. DLNRs on the scale of few lattice spacings can be clearly recognized (Fig. 2a). They are similar to domain patterns commonly observed in ferroelectrics[23]. From a previous study on bulk NBT-25ST[24], it is known that nanodomains cease to exist around 350 °C. Thus, the DLNRs observed at 800 °C cannot be attributed to a long-range ferroelectric spontaneous polarization. Moreover, one might assume that they are related to the presence of polar nanoregions (PNRs) in a relaxor state[25,26]. A recent quantitative analysis demonstrated that the fraction of PNRs in 0.94Na$_{1/2}$Bi$_{1/2}$TiO$_3$-0.06BaTiO$_3$ (NBT-06BT) is negligible above 700 °C[27]. A detailed quantification of the PNRs in NBT-25ST is beyond the scope of this work. However, by comparing the temperature-electric field phase diagrams of NBT-25ST and NBT-06BT[28], we may safely neglect the existence of PNRs above 700 °C in NBT-25ST due to its lower transition temperatures. The observed DLNRs are thus a consequence of other physical mechanisms.

We first investigated whether there is a homogeneous strain distribution within the particle. Elemental EDS mapping revealed that the synthesized NBT-25ST nanoparticles exhibit a core–shell structure. The Sr$^{2+}$ concentration increases towards the edge of the nanoparticles (Fig. 2b), while the other elements are homogeneously distributed (Fig. 2c–f). This is a direct consequence of the slow diffusion of the Sr$^{2+}$ in stoichiometric NBT-25ST[8]. A gradient of Sr$^{2+}$ leads to a chemically induced lattice strain because of the differences in ionic radii of the A-site cations[21]. This effect is usually referred as the Vegard effect[29,30]. In this case, the magnitude of the eigenstrain increases from the center to the edge. This eigenstrain influences the total strain distribution in the nanoparticles. As shown in recent work on strontium titanate, atomic-scale measurements of local displacements due to the flexoelectric effect have been reported[31]. However, for the NBT-25ST nanoparticle system, the measurement of atomic-displacements for the whole nanoparticle is nontrivial. Nevertheless, a quantitative assessment of the total strain distribution by atomic-displacement mapping in small regions of interest is possible (see Supplementary Figure 2), which indicates large strain gradients within the single nanoparticle. The value of strain ranges from −0.2% to 0.2 % within a distance of 3.8 nm. Figure 3a, b shows the same core–shell nanoparticle of Fig. 2 with a magnification of the DLNRs shown in Fig. 3c. The DLNRs highlighted in the zoom-in image could either be a result of the nonpolar lattice strain mismatch or originate from flexoelectricity as a consequence of the strain gradient.

**Flexoelectric-ferroelectric phase field modeling.** In order to clarify the origin of the DLNRs, we performed finite element flexoelectric-ferroelectric phase field simulations (Fig. 3d–f). The experimentally observed shape and size of the nanoparticle were implemented in the simulation. Open-circuit[32] boundary conditions are assumed for the freestanding nanoparticle without contacting the biasing electrodes. According to the Vegard law[30], the lattice parameter is linearly changed with the constituent's concentration. We treat the Vegard strain as the eigenstrain[33,34] in the phase field simulation. The Sr$^{2+}$ concentration is assumed to increase linearly from the center to the edge. Hence, the eigenstrain is set to increase from the center to the edge accordingly, as defined in Eq. (6) and visualized in Fig. 3d. Figure 3e shows the calculated polarization induced by the strain gradient. The polarization vector has its highest magnitude at the center of the nanoparticle and decreases toward the edges. Multiple polarization vortices are also observed within the particles. As seen in Fig. 3e, a distinct polarization configuration composed by a network of DLNRs is formed. The overall polarization configuration of the experiment and the simulation slightly differ, which may be due to the simplified strain distribution needed for the calculations. However, by comparison between the enlarged DLNRs observed in Fig. 3c and the polarization distribution shown in Fig. 3f, it is clear that they are analogous. In the phase field simulation, Landau energy coefficients[35] for the cubic phase were used, which indicates zero ferroelectric spontaneous polarization at the given temperature. Therefore, the DLNRs in the simulation can only be attributed to the high-order (gradient) coupling between mechanical strain and electric polarization. It should be noted that in the paraelectric phase, electrostriction still exists. The contribution of electrostriction to the polarization of the nanoparticles is two orders of magnitude lower as compared to the polarization induced by flexoelectricity (see Supplementary Figure 2–4). Hence, the electrostrictive effect is not responsible for the DLNRs and can be safely neglected. The phase field simulation shows that flexoelectricity can lead to the observed DLNRs.

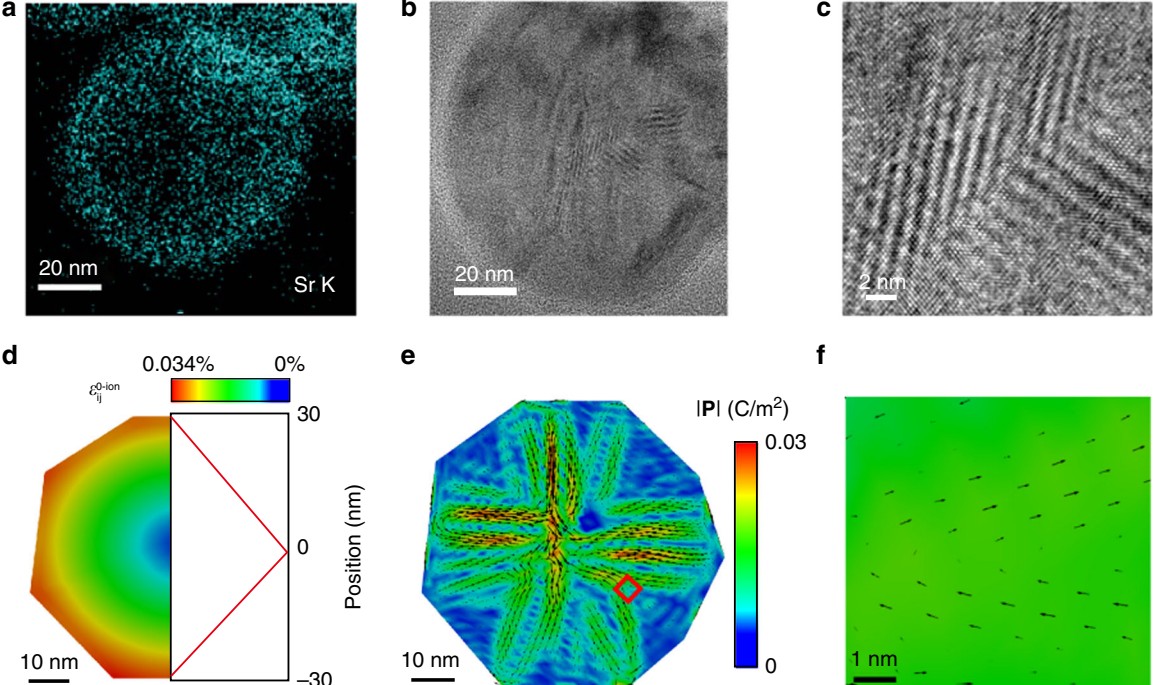

**Fig. 3** Experiment and simulation of domain-like nanoregions (DLNRs). **a** Energy-dispersive X-ray spectroscopy elemental map showing the $Sr^{2+}$ distribution. **b**, **c** Transmission electron microscopy (TEM) image of the core–shell nanoparticle and the enlargement of the red box region in **b**. **d** The linear distribution of the eigenstrain along the radius direction according to **a**. The symbol $\varepsilon_{ij}^{0-ion}$ stands for the eigenstrain due to the strontium inhomogeneity through the Vegard effect. **e** Flexoelectric-ferroelectric phase field simulation results of the polarization for the whole nanoparticle, experimentally shown in **b** The symbol |**P**| represents the magnitude of the polarization. **f** The enlargement of the red box region in **e** showing similar DLNRs as observed in the TEM images of **c**

To examine the origin of the DLNRs experimentally, electric field in situ TEM at 800 °C was performed. The core–shell structure of the in situ calcined NBT-25ST nanoparticle is revealed by the TEM image shown in Fig. 4a. The image was obtained along a $[113]_{pc}$ type zone axis at 0 kV mm$^{-1}$ and the coherency of core and shell can be observed in the corresponding Fast Fourier Transforms (FFTs). As seen in the magnified image in Fig. 4a, atomic resolution imaging was possible. Though not obvious in some regions in the shell, we observed DLNRs within the nanoparticle, in which the selected regions are magnified in Fig. 4d–f to aid visualization. The strain gradient across the core–shell interface denoted in Fig. 4a was quantified using a geometrical phase analysis (GPA) and strain distribution of the whole selected region can be found in Supplementary Figure 2. The core and shell DLNRs are in a stable configuration at the given conditions.

While keeping the temperature constant at 800 °C, an electric field was applied in a defined direction (black arrows), as seen in the TEM images shown in Fig. 4b, c. The images were Wiener filtered for noise reduction[36]. Figure 4g–i shows the corresponding FFTs of the core and shell areas, respectively. The bright spots observed in Fig. 4g–i are analogous to Bragg spots observed in conventional electron diffraction patterns[37]. They correspond to crystallographic planes visible in the high-resolution TEM images. Variation in the FFTs can be used to monitor the changes in the local crystal structure and related switching processes. The FFT patterns of the initial state (Fig. 4a) indicates that the particle is oriented along the $[113]_{pc}$ zone axis and features a pseudocubic crystal structure. The observed DLNRs are modified under the electric field shown in Fig. 4b, c. The white arrow in the FFT shown in Fig. 4i marks the electric field-induced splitting in the $(2\bar{2}0)$ reflex along the electric field. This indicates that there are two different polarities coexist, e.g., the dark and

bright regions as shown in Fig. 4d–f. The red arrow indicates a longer range ordering as expect for DLNRs. Changes in the domain-like configuration in the core and the shell become apparent at an electric field of 11.0 kV mm$^{-1}$ (Fig. 4b). Moreover, the formation of some DLNRs also occurs in the shell, as displayed in Fig. 4d–f. Further increasing the electric field to 21.9 kV mm$^{-1}$ (Fig. 4c) leads to more pronounced DLNRs. Besides, several DLNRs are nucleated within the shell.

In order to confirm the phenomena observed in Fig. 4a–c, phase field simulations were carried out by implementing the corresponding electric field applied to the particle (Fig. 4j–l). The black arrows inside the simulated nanoparticle indicate the direction of the local polarization and the color scheme shows the magnitude of the polarization vector. The initial polarization configuration at zero electric field (Fig. 4j) changes with increasing electric field. Coalescence of nanoregions is clearly observed as the electric field increases. The evolution of the polarization under electric field can be explained by the superposition of the initial flexoelectricity-induced polarization and the one induced by the electric field. When the electric field is high enough, the polarization induced by flexoelectricity is overshadowed. This is reflected in the evolution of the FFTs. Both the experimental and simulation results suggest that the DLNRs observed at extreme temperature are polar. This implies that the flexoelectric effect is the main reason for the formation of the DLNRs. The change of polarization and FFTs can be observed in the Supplementary Video 1 and prove that the process is fully reversible. The evolution and nucleation of DLNRs under bias electric field proves that a nonpolar lattice strain mismatch is not responsible for this phenomenon.

In order to critically assess our hypothesis, we synthesized samples with a different A-site doping ($Bi^{3+}$-deficient 75 mol% $Na_{1/2}Bi_{0.49}TiO_3$-25 mol% $SrTiO_3$) through a conventional solid-

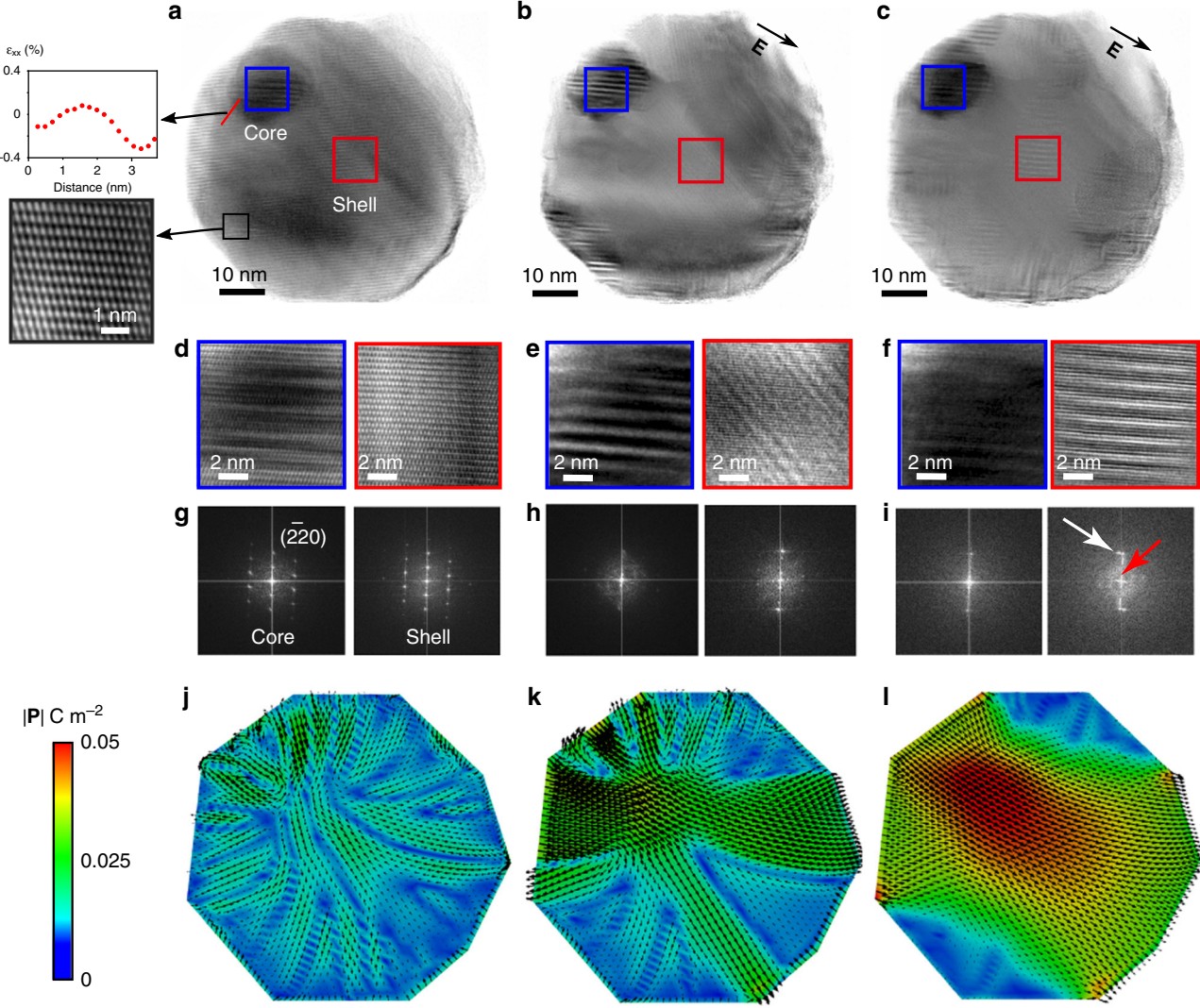

**Fig. 4** Demonstration of switchable flexoelectric-based polarization at extreme temperature. **a–c** Transmission electron microscopy (TEM) bright-field images taking along a [113]$_{pc}$ type zone axis, of a core–shell nanoparticle obtained at 800 °C with electric field of 0, 11.0, and 21.9 kV mm$^{-1}$, respectively. The magnified image of **a** shows an atomic resolution. The local strain distribution along the core–shell interface and the location of the profile is shown as a solid line in **a**. The black arrows in **b** and **c** indicate the direction of the electric field. Panels **d–f** are magnifications showing domain-like nanoregions (DLNRs) within the core and shell marked by the blue and red box in **a–c**. **g–i** The Fast Fourier Transforms (FFTs) of core and shell along a [113]$_{pc}$ type zone axis. The white arrow in **i** indicates an electric field-induced splitting in the ($2\bar{2}0$) reflex along the electric field indicating that there are two different polarities coexist. The red arrow in **i** indicates a longer range ordering as expect for DLNRs. **j–l** Phase field simulation of the domain patterns under the corresponding electric field. The color map indicates the polarization magnitude, while the black arrows the polarization vector. The symbol |**P**| in the legend represents the magnitude of the polarization

state route. According to our recent work[21], this stoichiometry should lead to a much more homogeneous Sr$^{2+}$ distribution within the nanoparticle rendering no core–shell structure and thus no noticeable strain gradient. The experimental setup and working environment were identical. Although some minor chemical heterogeneities can be observed (Supplementary Figure 7), the particles analyzed feature neither core–shell structure nor DLNRs. The lack of a long-range chemical gradient in Na$_{1/2}$Bi$_{0.49}$Ti-25ST results in negligible polarization (Supplementary Figure 13).

The role of oxygen vacancies under electric field should not be neglected. In recent work of Das et al.[38], controlled manipulation of oxygen vacancies in STO under mechanical loading was reported. In that case, flexoelectricity enabled the redistribution of oxygen vacancies. In our case, the oxygen vacancies may similarly influence the strain distribution and therefore the polarization by

the flexoelectric effect. However, the resistivity of NBT-25BT bulk samples is relatively high even at 800 °C[21] and it was not possible to measure the standard semi-circles expected in Nyquist plots. In situ TEM measurements to quantify the role of oxygen vacancies on flexoelectricity with atomic resolution at 800 °C falls out of the scope of the present article.

**Conclusions**. Through a combined experimental and phase field modeling approach, we observed nanoscale flexoelectricity at extreme temperature by tuning cation diffusion. NBT-25ST core–shell nanoparticles with a Sr$^{2+}$ chemical gradient were used as a model system to tune local strain by controlled ionic diffusion. The in situ synthesis route yielded core–shell nanoparticles that exhibit domain-like nanoregions DLNRs at 800 °C and were found to be in a stable configuration within the time

scale of the experiments. Electric field and temperature-dependent in situ TEM together with phase field flexoelectric simulations provide a detailed description and explanation of this phenomenon. Our results indicate that the polarization distribution is a direct consequence of nanoscale flexoelectricity and that it can be switched by applying an electric field inside a TEM. By comparing our results for samples with differing Bi deficiencies, we reaffirm the importance of a chemical gradient to generate strain and flexoelectric-induced polarization at high temperature. This contribution should motivate the study and development of other high-temperature flexoelectric nanomaterials.

## Methods

**Initial powders and in situ synthesis process.** The starting raw powders were produced via a mixed oxide route using reagent grade oxides and carbonates (Alfa Aesar GmbH, Karlsruhe, Germany). To achieve this, $Bi_2O_3$ (99.975%), $Na_2CO_3$ (99.5%), $TiO_2$ (99.9%), and $SrCO_3$ (99%) were mixed according to the $0.75Na_{1/2}Bi_{1/2}TiO_3$–$0.25SrTiO_3$ stoichiometric formula. The resulting uncalcined powder was dispersed in ethanol with an ultrasonic bath for 10 min. Drop-casting droplets of ultrasonically dispersed suspensions containing NBT-25ST milled powders on electro-thermal nano-chips (DENSsolutions, The Netherlands) was performed to investigate the particle synthesis and the functional properties of the core–shell nanoparticles. Given our previous knowledge on tuning cation diffusion of $Sr^{2+}$ through Bi stoichiometry, calcined Bi-deficient 75 mol% $Bi_{0.49}Na_{1/2}TiO_3$-25 mol% $SrTiO_3$ nanoparticles (with no core–shell) were synthesized following the synthesis steps described elsewhere[19]. One can find the temperature profile in Supplementary Figure 1. The Nano-Chips were individually calibrated and have a temperature accuracy of <5% and a temperature stability of <0.01 °C at 800 °C. The temperature of the nano-chip was gradually increased by 10 °C/min room temperature to 300 °C and held for 45 min. Subsequently, we increased the temperature to 600 °C for 120 min and a final step was done at 800 °C. After the full process $0.75Na_{1/2}Bi_{1/2}TiO_3$–$0.25SrTiO_3$ core–shell nanoparticles were formed.

**Electron microscopy and multiphysics simulation.** Transmission electron microscopy was performed using a JEOL JEM-ARM200F atomic resolution TEM (Tokyo, Japan) operated at 120 kV and an 80−300 FEI Titan microscope (Hillsboro, USA) equipped with an X-Max$^N$ 100TLE EDS-system (Oxford, UK). Energy-dispersive X-ray spectroscopy (EDS) was done with an Oxford EDS-system X-Max$^N$ 100TLE provided with a windowless 100 mm$^2$ sensor allowing for ultra-high solid angle acquisition. The detector delivers a high sensitivity for all elements, especially low energy X-rays. STEM-EDS mapping was performed with AztecTEM digital mapping software (Oxford, UK) and spatial drift correction was applied. Elemental maps were obtained with a resolution of 512 × 512 with a dwell time of 0.5 ms and a sweep count of 450. Combined heating and electrical biasing were carried out with a Lightning D9+ in situ TEM holder (DENSsolutions, The Netherlands). The heating conditions are established using a four-point-probe configuration, which excludes the influence of the cable's resistance, uncertain contact resistance and lead wires resistance, by using separate pairs of current-carrying and voltage-sensing electrodes to make more accurate measurements. Therefore, two electrodes supply a current to the microheater, which heats up through Joule heating, and the remaining two read out the resistance, which is then translated into temperature by means of the microheater's temperature coefficient of resistance (TCR). The latter, being a material property, defines the change in resistance as a function of the temperature. Consequently, the four-point-probe measurement in combination with the control unit results in a closed loop feedback system, which guarantees that despite potential thermal fluctuations inside the TEM column, the system will compensate for it to maintain the extreme temperature stability (millikelvin regime). The biasing conditions were simulated using a three-dimensional finite element analysis model (COMSOL Multiphysics) which includes the geometry of the heating-biasing chip and the nanoparticle diameter. An electric field of approximately 20 kV mm$^{-1}$ (100 V) can be applied. The relative permittivity of the nanoparticle for the simulation was set to 1500, which was found to be the corresponding value in earlier work[39]. The FFT images were generated by extracting regions-of-interest (ROI) of exactly the same position in both, the core and the shell regions of a series of TEM images.

**Flexoelectric-ferroelectric phase field simulation.** According to previous work[40,41], the total electrical entropy energy density of the ferroelectric system includes the bulk separation energy, gradient energy, elastic energy, electrostatic energy, electric–mechanical coupling energy with the addition of flexoelectric term:

$$H = H^{bulk} + H^{grad} + H^{ela} + H^{ele} + H^{coup} + H^{flexo} \tag{1}$$

Following the previously developed phase field ferroelectric models, the spontaneous polarization is taken as the order parameter. It allows explicit

formulation of the irreversible (spontaneous) and reversible (dielectric and piezoelectric) contribution of the electric displacement. The bulk free energy density is expressed up to the eighth-order term, i.e.:

$$H^{bulk} = \alpha_i P_i^2 + \alpha_{ij} P_i^2 P_j^2 + \alpha_{ijk} P_i^2 P_j^2 P_k^2 + \alpha_{ijkl} P_i^2 P_j^2 P_k^2 P_l^2 \tag{2}$$

where $\alpha_i$, $\alpha_{ij}$, $\alpha_{ijk}$, and $\alpha_{ijkl}$ are the Landau energy coefficient tensors and the Einstein summation notation is applied in the present paper. The values of these coefficients can be found in Supplementary Table 1.

The gradient energy density $H^{grad}(P_{i,j})$ is represented by the spatial derivatives of the polarization and takes the form:

$$H^{grad} = G_{ijkl} P_{i,j} P_{k,l} \tag{3}$$

where $G_{ijkl}$ is a fourth-order tensor. By assuming isotropy[42], it has three nontrivial-independent components, $G_{11}$, $G_{12}$, and $G_{44}$, where $G_{11} = G_{1111} = G_{2222}$, $G_{12} = G_{1122} = G_{2211}$, and $G_{44} = G_{1212} = G_{2121}$. The other components in the tensor are zero.

The elastic energy density can be expressed as:

$$H^{ela} = \frac{1}{2} c_{ijkl} \varepsilon_{ij}^{ela} \varepsilon_{kl}^{ela} = \frac{1}{2} c_{ijkl} \left( \varepsilon_{ij} - \varepsilon_{ij}^0 \right) \left( \varepsilon_{kl} - \varepsilon_{kl}^0 \right) \tag{4}$$

where $c_{ijkl}$ is the elastic stiffness tensor, $\varepsilon_{ij}^{ela}$ the elastic strain, $\varepsilon_{ij}$ the total strain and $\varepsilon_{ij}^0$ the non-elastic strain contribution. The non-elastic strain contains two terms in the simulation, i.e.:

$$\varepsilon_{ij}^0 = \varepsilon_{ij}^{0p}(P_i) + \varepsilon_{ij}^{0-ion}(\mathbf{x}) \tag{5}$$

where $\varepsilon_{ij}^{0p}(P_i)$ is the eigenstrain induced by the spontaneous polarization and $\varepsilon_{ij}^{0-ion}(\mathbf{x})$ is the eigenstrain induced from $Sr^{2+}$ concentration. Based on the energy-dispersive X-ray spectroscopy elemental mapping shown in Fig. 3, $Sr^{2+}$ concentration increases from the center to the boundary of the nanoparticle. For simulation, a linearly increase of $Sr^{2+}$ concentration is assumed. The resultant eigenstrain distribution is assumed to be:

$$\varepsilon_{ij}^{0-ion}(\mathbf{x}) = |\mathbf{x} - \mathbf{x}^{center}| W \delta_{ij} \tag{6}$$

where $W\delta_{ij}$ describes the isotropic mismatch strain induced by the ion and W takes a positive value since the radius of $Sr^{2+}$ is larger than those of $Bi^{3+}$ and $Na^+$. Here $\delta_{ij}$ is the Kronecker symbol. The symbols $\mathbf{x}$ and $\mathbf{x}^{center}$ are the position vector of the point under consideration and of the particle center, respectively.

The electrostatic contribution can be expressed as:

$$H^{ele} = -\frac{1}{2} k_{ij} E_i E_j - P_i E_i \tag{7}$$

where $k_{ij}$ is the dielectric tensor, and $E_i$ the electric field.

The electromechanical coupling energy density can be expressed as:

$$H^{coup} = \left( \varepsilon_{ij} - \varepsilon_{ij}^0 \right) e_{ijk}(P_l) E_k \tag{8}$$

where $e_{ijk}(P_l)$ is a third-order piezoelectric tensor which depends on the polarization.

The flexoelectric contribution is given as:

$$H^{flexo} = -\frac{1}{2} f_{ijkl} \left( P_i \varepsilon_{kl,j} - P_{i,j} \varepsilon_{kl} \right) \tag{9}$$

The flexocoupling coefficients have three nontrivial-independent components, $f_{11}$, $f_{12}$, and $f_{44}$, where $f_{11} = f_{1111} = f_{2222}$, $f_{12} = f_{1122} = f_{2211}$, and $f_{44} = f_{1212} = f_{2121}$. The flexocoupling coefficients $f_{11}$, $f_{12}$, and $f_{44}$ are set to 0.02, 0.7, and 0.3 V, respectively according to the work on strontium titanate by Zubko et al.[43] and Chen et al.[44].

The evolution of the polarization is described by the time-dependent Ginzburg–Landau equation:

$$\frac{\partial P_i}{\partial t} = -M \frac{\delta H}{\delta P_i} \tag{10}$$

where $M$ is the mobility parameter. For the mechanical equilibrium and charge conservation, the following equations should be fulfilled:

$$\sigma_{ij,j} - f_i = 0 \tag{11}$$

$$D_{i,i} = q \tag{12}$$

where $\sigma_{ij}$ and $D_i$ are stress and electric displacement, respectively. These two

quantities are calculated by:

$$\sigma_{ij} = \frac{\delta H}{\delta \varepsilon_{ij}} = c_{ijkl}\left(\varepsilon_{kl} - \varepsilon_{kl}^{0p} - \varepsilon_{kl}^{0-ion}\right) - b_{ijk}E_k + \frac{1}{2}f_{ijkl}\frac{\partial P_k}{\partial x_l} \tag{13}$$

$$D_i = -\frac{\delta H}{\delta E_i} = \kappa_{ij}E_j + b_{ijk}\left(\varepsilon_{jk} - \varepsilon_{jk}^{0p} - \varepsilon_{jk}^{0-ion}\right) + P_i. \tag{14}$$

Equations (10–14) are implemented by the finite element method in the Finite Element Analysis Program (FEAP)[45].

**Code availability**. The code is written as a user element in FEAP. The finite element framework can be found from the link: http://projects.ce.berkeley.edu/feap/. The user element can be provided upon request.

## Data availability

The data supporting the findings of this study are detailed in the paper and its supplementary information files.

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

## Acknowledgements

L.M.-L. acknowledges financial support from the European Union Seventh Framework Program under Grant Agreement 312483/ESTEEM2 (Integrated Infrastructure Initiative–I3) and the European Research Council (ERC) "Horizon 2020" Program under Grant No. 805359—FOXON. L.M.-L. and A.Z. acknowledge funding from the Deutsche Forschungsgemeinschaft (DFG) under research grant MO 3010/3-1. The JEOL JEM-ARM-F transmission electron microscope employed for this work was partially funded by the German Research Foundation (DFG/INST163/2951). S.W. and B.-X.X. acknowledge financial support by the "Excellence Initiative" of the German Federal and State Governments and the Graduate School of Computational Engineering at the Technische Universität Darmstadt and acknowledge the use of the Lichtenberg High Performance Computer. L.M.-L., M.Y. and B.-X.X acknowledge financial support from the Hessen State Ministry of Higher Education, Research and the Arts via LOEWE RESPONSE. M.A. acknowledges support from the Feodor Lynen Research Fellowship Program of the Alexander von Humboldt Foundation. Partial financial support of the Deutsche Forschungsgemeinschaft (DFG) Leibniz Program under RO954/22-1 was received. The authors thank U. Kunz and S. Steiner for assistance with TEM sample and powder preparation.

## Author contributions

L.M.-L., Q.X., and M.A. designed and performed the initial heating experiments. M.A. prepared the starting powders. L.M.-L., H.H. P.-G., R.G.S., and Y.P. designed and tested the electro-thermal chip. L.M.-L, Y.P., and A.Z. designed and performed the combined heating and electrical biasing experiments. H.H. P.-G., and R.G.S. performed the COMSOL Multiphysics simulations. S.W., M.Y., and B.-X.X. designed and performed the modeling and phase field simulations. All authors discussed extensively the results and commented on the manuscript. L.M.-L. and S.W. wrote the manuscript. L.M.-L. and M. A. coordinated this investigation.

## Additional information

**Competing interests:** The authors declare no competing interests.

