## [Peer Review File · Nature Communications]

Reviewers' comments:

Reviewer #1 (Remarks to the Author):

The authors study chemically induced strain gradient in nanoparticles with core shell structure. The structure is formed in situ in TEM. The authors claim that the flexoelectric polarization associated with the strain gradient is origin of the polar nanodomains well above the Curie temperature. These nanodomains are switchable by external electric field as indicated by in-situ TEM observations and phase-field modelling.

The work is interesting and results potentially significant. As it is, however, the paper is not clearly written and I have many questions which would have to be answered before I could consider paper for publication. I will give my comments in no particular order.

Page 1: use of term flexo-based " and similar ("flexo-induced"). "Flexo" itself relates to bending or flexing while flexoelectric effect as we understand it today relates generally to a strain gradient which is not necessarily obtained by bending. That is precisely the case in this paper. Therefore, I would avoid use of this term "flexo-xxx" and would simply say "flexoelectric based" or similar.

Page 1: The authors write "A key question remains though; can we design flexoelectric nanomaterials without the need for an external mechanical load?"

The authors then claim that they have accomplished this. This (having strain gradient without external forces) is in fact something that has been discussed a lot in the literature: see ref. 11 and several papers by Morozovska, Tagantsev, L. Martin (compositionally graded materials, ACS Nano, 2015), and Noheda (2011, NatMat), where they discuss flexoelectric effect which originates due to strain gradients that are built in the structure without applying an external field. Those authors discuss cases of strain gradients in thin films, due to compositional variation, around domain walls, and in general nonhomogeneous structures. The present work should be put properly in this earlier context and avoid claiming originality on this particular point.

Page 2: The authors write " From the simulations, we estimated that the electric field distribution within the core-shell BNT-25ST nanoparticle is homogenous." My understanding of Fig. 1 is that it show models of the electric field generated at sample holder, but that nanoparticle is not placed there. So, while the field at the sample holder is homogeneous I do not see how electric field inside a core-shell nanoparticle (and that is what matters) can be homogeneous. Composition, shape, dielectric constant will vary across an inhomogeneous particle. So this should be somehow commented in the paper (or explained why I am wrong in which case there is no need to change the text).

Page 2: The authors wrote " Thus, the domain like nanoregions observed at 800 °C cannot be attributed to spontaneous polarization, since this can only exist in non-centrosymmetric crystal systems." This statement is not correct. I am sure that the authors are familiar with existence of polar nano regions within paraelectric (centrosymmetric) phase of many ferroelectric or ferroelectric-like materials. These regions are sometimes called Burns' regions. See for example Gehring, 2009 PRB and Dkhil , 2009, PRB; and others.

Page 3: the authors wrote " Charge free boundary conditions are assumed for the freestanding nanoparticle without contacting the biasing electrodes." I am not sure this can be done. If such small particles contain polar regions the total polarization of the particle will not average out (due to asymmetry in shape) and some charges on the surface must appear if electrodes are not connected. This is particularly case for operation in TEM vacuum where there is no possibility for neutralizing surface charges from atmosphere. So, I do not see that the effects of depolarizing field can be neglected.

Page 4: the authors wrote " If the domain-like nanoregions observed in TEM were nonpolar, their

pattern should be insensitive to the applied electric field. On the other hand, if they were polar, the electric field should affect the pattern. " I do not think this is in general true. If a strain gradient is present in the sample (the main premise of the paper) then electric field may affect distribution of the strain (and therefore pattern of nanoregions) through electrostrictive effect if nanoregions have anisotropic shape.

Page 4: the authors wrote "fully reversible polarization changes in the core-shell BNT-25ST nanoparticle were observed (see Supplementary Video and Fig. S3)." Could authors explain more clearly to the readers which features in Fig. S3 and the video demonstrate polarization changes? How do authors see polarization in those figures?

I am also not sure why particle shown in Fig. S3 and Figure 3 has "core shell" structure? It looks more as an off center inclusion. I understand term "core shell" applied to Sr distribution in Figure 2. What is Sr distribution for the particle shown in Fig 3 and S3? Is it concentrated in the dark region (marked core?).

About Figure 4 and related text in the manuscript, the authors wrote "Agreements between the experimental and simulated domain patterns imply that the flexoelectric effect is the reason for the formation of the star-like polar nanoregions." Frankly, I am having difficulties seeing features in experimental pictures that correspond to simulated pictures in Fig. 4. The same holds for Fig. 3 Where in Fig 3a are the star-like features seen in Fig. 3c? Could the authors help the reader with this by, for example, outlining in Fig. 3a features that correspond to some features in Fig. 3c?

Page 5 and Figure S4: The authors wrote " The absence of the domain-like nanoregions in this case provides a strong evidence that, Sr²⁺ heterogeneity manifested as a core-shell structure is responsible for the strain gradient and the reason of flexo-induced polar nanoregions." What is the origin of contrast shown in figure S4? An untrained eye, which most of the readers would be, can interpret contrast in S4b (white regions) as some kind of nano regions. Also, the blue and red signals for Sr at 14keV are clearly different. Can authors show elemental map as they did in Fig. 2? As presented, these claims about absence of strain gradient in sample shown in Fig, S4 are not very convincing. Could authors show statistics by covering more regions of the grain and really showing that sample does not show strain gradient.

Why does not VBi cause strain gradient? Is it because this sample (nonstoichiometric) was calcined ex-situ? What if stoichiometric sample was calcined ex situ? Would it show strain gradient? Is in situ preparation a key condition for the gradient? If so, how reproducible this effect is?

Page 5: The authors wrote " The calculation shed lights on a new mechanism to minimize the domain size of ferroelectrics" and " Consequently, a small strain gradient can produce such star-like polar nanoregions. Producing a strain gradient near phase transitions in ferroelectrics can thus be used as a strategy to minimize domain size and concomitantly maximize electromechanical properties ".

These statements are very misleading and far-reaching without any foundation. Observation of small domains in this work (whose ferroelectric character has not been proven) at high temperatures in in-situ prepared samples does not imply that other ferroelectric nano particles with a strain gradient would exhibit small domains. Do authors know if in-situ prepared nanoparticles (used in this work) would be ferroelectric at room temperatures? Are they ferroelectric at 800°C? Would the same small domain-size be present at room temperature? There is no general theory that shows that small domain size must lead to large response. That is a conjecture that has been often used in the literature (in some cases materials with a small domain size indeed show large response) but there are opposite examples (Jin, 2009, APL). I suggest to the authors to simply drop those claims as they are not central to their paper.

In conclusion section, the authors wrote " Electric field and temperature dependent in situ TEM experiments demonstrated that the domain-like nanoregions are polar and that they can be

switched."

Could authors explain better how experimental results demonstrated polar character of nano regions? Is it Figure 4 and how exactly?

Reviewer #2 (Remarks to the Author):

In their paper Dr. L. Molina-Luna et al., have claimed to realize switchable electrical polarization assisted by flexoelectricity in BNT-25ST core-shell nanoparticles. Flexoelectricity is a rapidly developing field, but so far, the center focus was exploring means of generating large nanoscale strain-gradient/polarization (non-switchable) and its applications. The successful demonstration of "switchable flexoelectric polarization", therefore, will be an important advancement in this field. However, the experimental evidence and arguments presented in the current manuscript are not strong enough to support the authors' claims.

1) According to their phase-field simulations, a strain-gradient of the order of 10^5 m^{-1} is necessary to stabilize domain-like polar nanoregions (in Fig. 3a). However, authors just provided a qualitative assessment of the strain-distribution based on the GPA analysis (Fig. S2). Thus, the authors could attempt to quantify the strain-gradient within the nanoparticles

2) Figures 4, S3, S5, and the supplementary video are most relevant in understanding the central claim made by the authors. I have following questions regarding these data.

a) Do I understand correctly, the "field-induced domain-like nanoregions" (line # 127) are the same as those induced by the flexoelectric effect in Fig. 3a? If yes, why the reflex corresponding to the domain-like nanoregions only appears in the FFT image obtained under an electric field of 11 KV/mm but not in the image obtained under the zero electric field?

b) In line# 131 authors claimed, "Under the removal of the electric field, the domain-like nanoregions vanish." Does it mean the polar nanoregions are unstable against electrical-cycling? If yes, why they are unstable?

c) Based on the phase-field simulations, authors argued that for high field strength, the electric field-induced polarization overshadows the flexoelectric-polarization. This results in the mere reorientation of polarization inside the nanoparticle. If only the polarization is reorienting, why the FFT images (in Fig. S3 and Supplementary video) corresponding to $E = -21.9 \text{ KV/mm}$ and $+21.9 \text{ KV/mm}$ look different? In the former case, alongside the primary reflexes, additional secondary reflexes can be observed. These secondary reflexes, however, are absent in the latter case. It will help if the authors clarify the origin and the relevance of this difference in the switching of polarization.

d) While discussing the effect of electrical biasing, authors do not consider the motion charged defects such as oxygen vacancies, which will be inevitably present in the nanoparticle given the inhomogeneous distribution of Sr^{2+} . At the operating temperature (8000 C), the oxygen vacancies would be highly mobile, and electrical biasing would readily alter their spatial distribution. This could affect the strain-distribution and hence the flexoelectric effect inside the nanoparticle. Thus, authors should clarify the role of ionic motion in the electric field-induced polarization switching.

e) Another important issue the authors have not discussed concerns the microscopic mechanism behind the electrical switching of polarization. In conventional ferroelectrics, the switchable polarization is due to the reversible ionic displacement under the applied electric field. In terms of the Landau energy landscape, this would correspond to switching between the two minima of the double well potential (Fig. S5). The authors showed in Fig. S5 that with increasing temperature the energy landscape changes from the double well into a single well. Therefore, it is not clear how the

polarization switches in the absence of two potential minima?

Reviewer #3 (Remarks to the Author):

In this manuscript the authors reported domain-like nanopolar regions in NBT-25ST core shell nanoparticles at elevated temperature when ferroelectric polarization usually vanishes, and attribute this polarization to the flexoelectric effect that arise from the diffusion of Sr²⁺ cations. The authors used both in-situ TEM characterization method and phase-field simulation to prove their assumptions. However I found the following questions critical that impede my recommendation for its publication in nature communications.

(1) The authors observed Sr²⁺ cations diffusion from center to the edges of the nanoparticles in TEM, and interpret it to a linearly increasing isotropic eigenstrain in the model. To my knowledge this strain is called chemical Vegard strain and the effect is called Vegard's effect. I think here the authors confuse Vegard chemical strain (which is due to the Vegard expansion of the lattice caused by mobile species) with the flexoelectric strain (which is induced from the polarization gradient $\epsilon_{ij} = F_{ijkl} * dP_k/dx_l$ via converse flexoelectric effect, see Eq. (13) in their own manuscript, also refer to paper PHYSICAL REVIEW B 83, 195313 (2011)). Therefore based on their modeling, the ferroelectric polarization at elevated temperature is actually induced by Vegard's strain effect, NOT flexoelectric effect.

(2) In Figure S5, the authors studied different domain patterns by tuning the eigenstrain via coefficient W (Vegard expansion coefficients), while in Eq. (9) the author actually introduced flexoelectric energy H_{flexo} to the total free energy. If the domain-like pattern at 800 oC is induced from flexoelectric effect as they argued, the authors should turn on and off flexoelectric contribution by setting $f_{11}=f_{12}=f_{44}=0 \sim 10V$ to see the differences in domain pattern, rather than tuning the Vegard expansion coefficient (W). I believe this can clarify which effect is responsible for the observed behavior.

(3) The material of study in this paper is 0.75Bi_{1/2}Na_{1/2}TiO₃-0.25SrTiO₃, However the Landau energy coefficients (α) used in the phase-field simulation are collected from the paper for BaTiO₃, "Li, Y. L., Cross, L. E. & Chen, L. Q. A phenomenological thermodynamic potential for BaTiO₃ single crystals. J. Appl. Phys. 98, 64101 (2005)." Although the authors did modify the temperature dependent coefficients α_1 based on different Curie temperature (813K instead of 115oC), the Curie constant and the other Landau energy coefficients are exactly the same. That probably explain why the calculated polarization in Fig. 3 and Fig. S5 are up to 0.26C/m², which is the reported spontaneous polarization of BaTiO₃, rather than 0.38C/m² for BNT-ST. Therefore I doubt if the modeling results based on the BTO coefficients can explain the real experimental observations in BNT-ST.

(4) The authors argued that the center of the nanoparticle is under compressive strain and the edge under tensile strain when the Sr diffuse from the center to the edges, which is counter-intuitive to me. Shouldn't more Sr ions at the edges induce compressive strain and the loss of Sr ions at the center induce tensile strain?

(5) In Figure 3(c) and Figure S5 (lower-right one), the scale bars do not match each other.

Based on these concerns, I cannot recommend it for publication in nature communications at its current form.

Reviewer #1 (Remarks to the Author):

The authors study chemically induced strain gradient in nanoparticles with core shell structure. The structure is formed in situ in TEM. The authors claim that the flexoelectric polarization associated with the strain gradient is origin of the polar nanodomains well above the Curie temperature. These nanodomains are switchable by external electric field as indicated by in-situ TEM observations and phase-field modelling.

The work is interesting and results potentially significant. As it is, however, the paper is not clearly written and I have many questions which would have to be answered before I could consider paper for publication. I will give my comments in no particular order.

We thank the reviewer for his/her recognition of the potential significance of this work.

Page 1: use of term flexo-based " and similar ("flexo-induced"). "Flexo" itself relates to bending or flexing while flexoelectric effect as we understand it today relates generally to a strain gradient which is not necessarily obtained by bending. That is precisely the case in this paper. Therefore, I would avoid use of this term "flexo-xxx" and would simply say "flexoelectric based" or similar.

We have corrected the terminology accordingly. We also changed the "*Flexo-ferroelectric phase field simulation*" terminology to "*Flexoelectric-ferroelectric phase field simulation*" throughout the article. We have highlighted all corrected sentences in the revised manuscript.

Page 1: The authors write "A key question remains though; can we design flexoelectric nanomaterials without the need for an external mechanical load?" The authors then claim that they have accomplished this. This (having strain gradient without external forces) is in fact something that has been discussed a lot in the literature: see ref. 11 and several papers by Morozovska, Tagantsev, L. Martin (compositionally graded materials, ACS Nano, 2015), and Noheda (2011, NatMat), where they discuss flexoelectric effect which originates due to strain gradients that are built in the structure without applying an external field. Those authors discuss cases of strain gradients in thin films, due to compositional variation, around domain walls, and in general nonhomogeneous structures. The present work should be put properly in this earlier context and avoid claiming originality on this particular point.

We thank the reviewer for emphasizing the existence of important previous work that was not properly cited. We have now introduced more accurately the novelty of the work and discussed carefully the literature on the topic. The updated sentences are given below:

"Designing flexoelectric nanomaterials without the need for an external mechanical load has been previously reported by developing compositionally graded materials^{20,21} and engineering domain walls in ferroelectric thin

films²². Here, we provide a simple and effective route to design nanomaterials with a built-in strain gradient by controlling ionic diffusion²³. By making use of the strain gradient and the flexoelectric effect it was possible to produce nanoparticles with switchable polarization at 800 °C."

(in page 2)

Page 2: The authors write "From the simulations, we estimated that the electric field distribution within the core-shell BNT-25ST nanoparticle is homogenous." My understanding of Fig. 1 is that it show models of the electric field generated at sample holder, but that nanoparticle is not placed there. So, while the field at the sample holder is homogeneous I do not see how electric field inside a core-shell nanoparticle (and that is what matters) can be homogeneous. Composition, shape, dielectric constant will vary across an inhomogeneous particle. So this should be somehow commented in the paper (or explained why I am wrong in which case there is no need to change the text).

We appreciate this important comment. Indeed, we were referring to the electric field at the sample holder and certainly not within the core-shell nanoparticle. We changed the sentence to avoid misunderstandings from:

"From the simulations, we estimated that the electric field distribution within the core-shell BNT-25ST nanoparticle is homogenous."

to

"Fig. 1b and 1d show that the temperature and the applied electric between the electrodes is homogenous at the sample region."

(in Page 3)

In our phase field simulations, the influence of the composition, shape, and dielectric constant on the electric field and polarization distribution within the nanoparticle is considered.

Page 2: The authors wrote "Thus, the domain like nanoregions observed at 800 °C cannot be attributed to spontaneous polarization, since this can only exist in non-centrosymmetric crystal systems." This statement is not correct. I am sure that the authors are familiar with existence of polar nano regions within paraelectric (centrosymmetric) phase of many ferroelectric or ferroelectric-like materials. These regions are sometimes called Burns' regions. See for example Gehring, 2009 PRB and Dkhil , 2009, PRB; and others.

We agree that this sentence does not clarify the role of relaxors with PNRs. We also realized that we should clarify better the terminology used for the mechanisms that can lead to local polarization.

The potential origins of local polarization relevant to this work are:

- Spontaneous polarization in a ferroelectric state which can be described by the Landau polynomial.

- field induced polarization (ϵE).
- strain induced polarization (piezoelectricity and electrostriction).
- strain-gradient induced polarization (flexoelectricity).
- local spontaneous polarization in a relaxor state (i.e., PNRs in a relaxor state).

PNRs can exist in pseudo centrosymmetric crystal systems e.g. “polar clusters” near the local Curie temperature (T^*) as shown in the references of Gehring *et al.* and Dkhil *et al.* kindly provided by the reviewer. It is important to highlight though, that the Burns temperature (T_B) is generally defined as the temperature at which the PNRs vanish (Bokov, A. A. and Z-G. Ye. Recent progress in relaxor ferroelectrics with perovskite structure. (*Frontiers of Ferroelectricity*. Springer US, 31-52 (2006)). Thus, we do not expect any PNRs contribution to the polarization above T_B .

We probed the nanoparticles at 800 °C, which according to our estimation should be well above the T_B of NBT-25ST. By comparing the E-T phase diagram of NBT-25ST with NBT-6BT (Weyland *et al.*, *J. Mater. Sci.* **53**, 9393–9400 (2018)), it can be observed that NBT-25ST has lower transition temperatures than NBT-06BT. Vögler *et al.* demonstrated experimentally (*Phys. Rev. B*, **95**, 024104 (2017)) that the volume fraction of the PNRs tends to zero at 700 °C. Since NBT-25ST has lower E-T transition temperatures, we can safely conclude that the observed domain-like nanoregions do not originate from PNRs in a relaxor state, but rather are related to other physical mechanisms (e.g., flexoelectricity).

To avoid misunderstandings, in page 3 we changed the sentence from:

"Thus, the domain-like nanoregions observed at 800 °C cannot be attributed to spontaneous polarization, since this can only exist in non-centrosymmetric crystal systems."

to:

"From a previous study on bulk NBT-25ST²⁷, it is known that nanodomains cease to exist around 350 °C. Thus, the DLNRs observed at 800 °C cannot be attributed to a long-range ferroelectric spontaneous polarization. Moreover, one might assume that they are related to the presence of polar nanoregions (PNRs) in a relaxor state^{28,29}. A recent quantitative analysis demonstrated that the fraction of PNRs on 0.94Na_{1/2}Bi_{1/2}TiO₃-0.06BaTiO₃ (NBT-06BT) is negligible above 700 °C³⁰. A detailed quantification of the PNRs in NBT-25ST is beyond the scope of this work. However, by comparing the temperature-electric field phase diagrams of NBT-25ST and NBT-06BT³¹, we may safely neglect the existence of PNRs above 700 °C in NBT-25ST due to its lower transition temperatures. The observed DLNRs are thus a consequence of other physical mechanisms."

Page 3: the authors wrote "Charge free boundary conditions are assumed for the

freestanding nanoparticle without contacting the biasing electrodes." I am not sure this can be done. If such small particles contain polar regions the total polarization of the particle will not average out (due to asymmetry in shape) and some charges on the surface must appear if electrodes are not connected. This is particularly case for operation in TEM vacuum where there is no possibility for neutralizing surface charges from atmosphere. So, I do not see that the effects of depolarizing field can be neglected.

We agree that the terminology "charge free" is misleading. Actually, what we used is the open-circuit boundary condition, i.e., $D_{in_i} = 0$. Thereby the depolarizing field is taken into account. Due to this reason, the polarization distribution at the surface, as shown in Fig. 3e is rather different from that in the core region.

We change the sentence from:

"Charge free boundary conditions are assumed for the freestanding nanoparticle without contacting the biasing electrodes."

to:

"Open-circuit³⁴ boundary conditions are assumed for the freestanding nanoparticle without contacting the biasing electrodes."

(in Page 3)

Page 4: the authors wrote "If the domain-like nanoregions observed in TEM were nonpolar, their pattern should be insensitive to the applied electric field. On the other hand, if they were polar, the electric field should affect the pattern." I do not think this is in general true. If a strain gradient is present in the sample (the main premise of the paper) then electric field may affect distribution of the strain (and therefore pattern of nanoregions) through electrostrictive effect if nanoregions have anisotropic shape."

Firstly, we will discuss the role of the observed splitting of the $(\bar{2}\bar{2}0)$ reflex as an indicator for polarity.

We identified two main features in the FFTs that have different origins. (See updated Fig. 4c). The red arrow indicates a longer range ordering as expected for domain-like nanoregions. The white arrow marks the splitting of the $(\bar{2}\bar{2}0)$ reflex along the electric field. The splitting occurs due to the existence of two different polarized unit cell types. The same phenomenon is found in our observation of a single crystalline ferroelectric (Fig. 5 in Liu, H. *et al. in IEEE Transactions on Ultrasonics, Ferroelectrics, and Frequency Control* (2018).). Thus, it confirms that the observed DLNRs possess a polar character.

We would also like to mention that the contrast observed in the TEM images is neither from thickness fringes nor from Moiré patterns. Thickness fringes can be excluded due to the spherical nature of the particle. The spherical geometry would determine the shape of these fringes to be concentric following the

projected shape of the nanoparticle. This is not observed. Moiré patterns can also be excluded due to the coherent nature of the nanoparticle (See FFTs).

In order to clarify this point in the manuscript we added the following text to describe the contrast change under the electric field:

"...(FFT) correspond to crystallographic planes visible in the high-resolution TEM images. Variation in the FFTs can be used to monitor the changes in the local crystal structure and related switching processes. The FFT patterns of the initial state (Fig. 4a) indicates that the particle is oriented along the $[113]_{pc}$ zone axis and features a pseudocubic crystal structure. The observed DLNRs are modified under the electric field shown in Fig 4 b-c. The white arrow in the FFT inset of Fig. 4c marks the electric field induced splitting in the $(2\bar{2}0)$ reflex along the electric field. This indicates that there are two different polarities prevalent. The red arrow indicates a longer range ordering as expect for DLNRs."

(Page 4-5)

Secondly, we address whether electrostriction plays a role in the formation of domain-like nanoregions.

We agree that in this situation, the electrostriction exists and has an influence on the strain/polarization state. However, in the following, we show that the electrostriction-induced polarization is negligible as compared to flexoelectricity. At nanoscale, the strain gradient effect is more predominant than the strain effect.

Here, we use the finite element simulation to calculate the distribution of the polarization with only electrostriction considered. The total energy is given in the following form:

$$H(\varepsilon_{ij}, E_i) = \frac{1}{2} C_{ijkl} (\varepsilon_{ij} - \varepsilon_{ij}^0) (\varepsilon_{kl} - \varepsilon_{kl}^0) \quad (1)$$

where C_{ijkl} , ε_{ij} and ε_{ij}^0 are elastic stiffness, total strain and eigenstrain respectively. The eigenstrain is quadratically related to the field,

$$\varepsilon_{ij}^0 = \frac{\kappa^2}{2} Q_{ijkl} E_k E_l \quad (2)$$

where κ is the material permittivity, Q_{ijkl} is the electrostrictive coefficient and E_i is the electric field. From the energy function, constitutive relations can be obtained as:

$$\sigma_{ij} = \frac{\partial H}{\partial \varepsilon_{ij}} = C_{ijkl} (\varepsilon_{ij} - \varepsilon_{ij}^0) \quad (3)$$

$$D_i = -\frac{\partial H}{\partial E_i} = \kappa E_i + \kappa^2 Q_{ijkl} E_j \sigma_{kl} \quad (4)$$

The electrostriction-related polarization is

$$P_i^0 = \kappa^2 Q_{ijkl} E_j \sigma_{kl} \quad (5)$$

We gave the following input values of $Q_{11} = 0.02 \text{ m}^4/\text{C}^2$, $Q_{12} = -0.005 \text{ m}^4/\text{C}^2$ and $Q_{44} = 0.01 \text{ m}^4/\text{C}^2$ for the core and doubled for the shell (these values are typically found for NBT-based material. See ref. Zhang *et al. Adv. Mater.* **21**(46), 2009). The permittivity of the core-shell nanoparticle was 1000 times the value of the vacuum. We set the electric field to 20 kV/mm by providing this voltage difference between the two boundaries.

Shown in figure R1 are the corresponding potential and electric field distributions. Figure R2 shows the distribution of the electrostrictive-related strain. We can find that some relatively high strain region can only be found at the corner of the particle and the interface between the core and the shell. The magnitude of the strain here is 1% of the strain induced by the Sr^{2+} chemical gradient shown in the manuscript. Figure R3 gives the electrostriction-induced polarization P_i^0 . The color indicates the magnitude of the polarization and the arrow shows its direction. We found the magnitude of the polarization is below 0.0002 C/m^2 , which is $\sim 1.5\%$ of the flexoelectric-based polarization (0.03 C/m^2). The large flexoelectric-based polarization is due to the size effect: as the scale down to nanometers, the strain gradient is much higher, thus the flexoelectric effect is more predominate.

We agree that electric field may affect distribution of the strain (and therefore pattern of nanoregions) through electrostrictive effect if nanoregions have anisotropic shape, but this effect is not comparable with flexoelectric effect.

In order to clarify the electrostriction contribution, we added the following text:

"It should be noted that in the paraelectric phase, electrostriction still exists. The contribution of electrostriction to the polarization of the nanoparticles is two orders of magnitude lower as compared to the polarization induced by flexoelectricity (see Supplementary file; Fig. R4). Hence, the electrostrictive effect is not responsible for the DLNRs and can be safely neglected. The phase field simulation shows that flexoelectricity can lead to the observed DLNRs."

(Page 4)

Figure R1. **a** Potential and **b** electric field distribution of the nanoparticle.

Figure R2. Electrostrictive-related eigenstrain distribution.

Figure R3. Electrostriction-induced polarization distribution.

Page 4: the authors wrote "fully reversible polarization changes in the core-shell BNT-25ST nanoparticle were observed (see Supplementary Video and Fig. S3)." Could authors explain more clearly to the readers which features in Fig. S3 and the video demonstrate polarization changes? How do authors see polarization in those figures?

As previously discussed, the observed splitting of the $(2\bar{2}0)$ reflex is the indicator for polarity.

Please refer to the response for the previous question raised.

I am also not sure why particle shown in Fig. S3 and Figure 3 has "core shell" structure?

It looks more as an off center inclusion. I understand term "core shell" applied to

Sr distribution in Figure 2. What is Sr distribution for the particle shown in Fig 3 and S3? Is it concentrated in the dark region (marked core?).

Firstly, we would like to emphasize that the core does not necessarily need to be exactly in the center of the nanoparticle (for instance Fig. 7 in Acosta, M., *et al. J. Am. Ceram. Soc.*, 98, 3405-3422. (2015)). This depends on a diffusion-limited formation mechanism as discussed elsewhere (see Ref. Koruza, J. *et al J. Eur. Ceram. Soc.* 36, 1009–1016 (2016)).

Secondly, our experiments show that the core always features a lower Sr²⁺ content. Fig. 3 and S3 display the same particle as the supplementary video 1 for comparison. Our observation of the DLNRs indicates that the core and shell can only be of coherent nature (see FFTs of figure 4a-c and discussion above). The FFTs also reflect the perfect alignment of core and shell which would not be the case for an inclusion.

We included the following text in the manuscript that describes with much more detail the observations:

"To examine the origin of the DLNRs experimentally, electric field in situ TEM at 800 °C was performed. The core-shell structure of the in situ calcined NBT-25ST nanoparticle is revealed by the TEM image shown in Fig. 4a. The image was obtained along a [113]_{pc} type zone axis at 0 kV/mm and the coherency of core and shell can be observed in the corresponding Fast Fourier Transforms (FFTs). As seen in the left inset of Fig. 4a, atomic resolution imaging was possible. We observed DLNRs within the core and shell regions, which are magnified to aid visualization in the top insets. The strain gradient across the core-shell interface denoted in Fig. 4a was quantified using a geometrical phase analysis (GPA) in Supplementary Fig. S2. The core and shell DLNRs are in a stable configuration at the given conditions.

While keeping the temperature constant at 800 °C, an electric field was applied in a defined direction (black arrows), as seen in the TEM images shown in Fig. 4b-c. The images were Wiener filtered for noise reduction³⁷. The bottom insets show the corresponding FFTs of the core and shell areas, respectively. The bright spots observed in the FFT insets are analogous to Bragg spots observed in conventional electron diffraction patterns. They correspond to crystallographic planes visible in the high-resolution TEM (HRTEM) images. Variation in the FFTs can be used to monitor the changes in the local crystal structure and related switching processes. The FFT patterns of the initial state (Fig. 4a) indicates that the particle is oriented along the [113]_{pc} zone axis and features a pseudocubic crystal structure. The observed DLNRs are modified under the electric field shown in Fig 4 b-c. The white arrow in the FFT inset of Fig. 4c indicates an electric field induced splitting in the (2 $\bar{2}$ 0) reflex along the electric field. This indicates that there are two different polarities prevalent. The red arrow indicates a longer range ordering as expect for DLNRs. Changes in the domain-like configuration in the core and the shell become apparent at an electric field of 11.0 kV/mm (Fig. 4b). Moreover, the formation of some DLNRs also occurs in the shell, as displayed in the top inset. Further increasing the

electric field to 21.9 kV/mm (Fig. 4c) leads to more pronounced DLNRs, as displayed in the top insets of core and shell. Besides, several DLNRs are nucleated within the shell."

(page 4-5)

About Figure 4 and related text in the manuscript, the authors wrote "Agreements between the experimental and simulated domain patterns imply that the flexoelectric effect is the reason for the formation of the star-like polar nanoregions." Frankly, I am having difficulties seeing features in experimental pictures that correspond to simulated pictures in Fig. 4. The same holds for Fig. 3 Where in Fig 3a are the star-like features seen in Fig. 3c? Could the authors help the reader with this by, for example, outlining in Fig. 3a features that correspond to some features in Fig. 3c?

The terminology we used created a confusion that the reviewer properly highlights. In the experiments and calculations, we observe a distribution of domain-like nanoregions. We have updated Fig. 3 to compare the TEM observations and simulated polarization within an enlargement region. And we also provided a new image Fig. S8 in supplementary in which we marked the domain-like nanoregions of Fig. 3. From these comparisons, we can conclude that the observed DLNRs are similar to the simulated polarization distribution in small regions.

We agree that the "star-like" terminology is misleading and thus, we have eliminated it. We modified Fig 3 and the text accordingly to address the similarities between the experiment and the simulation:

"As seen in Fig. 3e, a distinct polarization configuration composed by a network of DLNRs is formed. The overall polarization configuration of the experiment and the simulation slightly differ, which may be due to the simplified strain distribution needed for the calculations."

Page 5 and Figure S4: The authors wrote "The absence of the domain-like nanoregions in this case provides a strong evidence that, Sr²⁺ heterogeneity manifested as a core-shell structure is responsible for the strain gradient and the reason of flexo-induced polar nanoregions." What is the origin of contrast shown in figure S4? An untrained eye, which most of the readers would be, can interpret contrast in S4b (white regions) as some kind of nanoregions.

Also, the blue and red signals for Sr at 14keV are clearly different. Can authors show elemental map as they did in Fig. 2? As presented, these claims about absence of strain gradient in sample shown in Fig. S4 are not very convincing. Could authors show statistics by covering more regions of the grain and really showing that sample does not show strain gradient.

We have now included an elemental map of Fig. S4 (as done in Fig. 2) for a larger region of the nanoparticle to include more statistics. It is shown in the supplementary Fig. S4c. The imaging mode used for Fig. S4a-b was HAADF-STEM imaging which is sensitive to atomic Z-contrast in combination with

energy-dispersive X-ray spectroscopy (EDS). The box region marked in red in Fig. S4b is a map of the mentioned region after sintering, color code as denoted in the legend). For each pixel there is an EDS spectrum as shown in Fig. S4d. The image shows intensities for selected characteristic X-ray peak intensities for the given species and thus, the map reveals compositional variations in the nanoparticle. The heating-step inside the transmission electron microscope did not result in a core-shell structure with a defined strain gradient in contrast to the stoichiometric NBT-25ST. We still resolve nanometer-scale segregated areas (appearing bright in S4b) but no defined chemical gradient is present. The difference in the EDS spectra are indeed induced by the different chemical compositions of the selected regions (brighter and darker in image intensity).

In order to corroborate whether the local chemical fluctuations of the nanoparticle can lead to the formation of domain-like nanoregions at 800 °C, we performed a new simulation that considers the ferroelectric and flexoelectric contributions to the polarization in a nanoparticle with a random eigenstrain distribution. The random eigenstrain distribution is set to mimic the presence of local chemical fluctuations appearing in Fig. S4b. Except for the eigenstrain distribution, all the other parameters are exactly the same as in the simulation shown in Fig. 3 in the manuscript. The random distributed eigenstrain is given by:

$$\varepsilon^0 = \begin{pmatrix} z & 0 \\ 0 & z \end{pmatrix}$$

where z is a random number in each simulated element, with the standard deviation of 2×10^{-5} . The real strain distribution is shown in Fig. R4a-c. The corresponding polarization distribution is shown in Fig. R4d. Note that the scale bar of Fig. R4d is the same as in Fig. 3e for comparison. The magnitude of the polarization is two orders of magnitude lower than the one calculated in Fig. 3c. Considering that all the other conditions are the same, it can be concluded that the radial strain distribution (ascribed to the Sr^{2+} chemical gradient) and its coupling with the flexoelectric effect are responsible for the domain-like nanoregions shown in Fig. 3e at 800 °C.

We added one paragraph in the main manuscript to highlight these aspects:

"In order to critically assess our hypothesis, we synthesized samples with a different A-site doping (Bi^{3+} -deficient 75 mol % $\text{Na}_{1/2}\text{Bi}_{0.49}\text{TiO}_3$ -25 mol % SrTiO_3) through a conventional solid-state route. According to our recent work²⁰, this stoichiometry should lead to a much more homogeneous Sr^{2+} distribution within the nanoparticle rendering no core-shell structure and thus no noticeable strain gradient. The experimental setup and working environment were identical. Although some minor chemical heterogeneities can be observed (Fig. S4), the particles analyzed featured neither core-shell structure nor DLNRs. The lack of long-range chemical gradient in this $\text{Na}_{1/2}\text{Bi}_{0.49}\text{Ti}$ -25ST results in negligible polarization (See supplementary response Fig. R4d)".

(page 6)

Figure R4. **a-c**, Strain distribution with random distributed eigenstrain. **d**, The corresponding polarization distribution based on the ferroelectric-flexoelectric model.

Why does not VBi cause strain gradient? Is it because this sample (nonstoichiometric) was calcined ex-situ? What if stoichiometric sample was calcined ex situ? Would it show strain gradient?

We thank the reviewer for pointing this out. VBi should have a certain effect on the strain distribution, but our calculations and measurements indicate that the main role is imposed by the Sr^{2+} chemical distribution. The strain gradient that we observe does not originate from point defects since it has a larger range behavior (please see discussion above).

From our understanding, the importance of the VBi is not to determine the strain gradient, but we use it as a mean to control the cation diffusion of Sr^{2+} (see our recent article Frömling *et al. J. Mater. Chem. C*, **6** 738 (2018)). The material with a core-shell is stoichiometric. The observation of a core-shell structure in the nanoparticles is a result of this compositional stoichiometry. The core-shell (meaning Sr^{2+} heterogeneity) generates the strain gradient.

Is in situ preparation a key condition for the gradient? If so, how reproducible this effect is?

In situ preparation of the sample is by no means a necessary condition to generate core-shell nanoparticles. We demonstrate the formation of core-shell nanoparticles produced ex-situ, which are analogous to the ones synthesized here

in situ, in previous manuscripts (Acosta *et al.*, *J. Am. Ceram. Soc.* **98** 3405 (2015) and Koruza *et al.*, *J. Eur. Ceram. Soc.* **36** 1009 (2016)). We follow in this work the formation of the core-shell in situ to evaluate the strain and nanodomains formation accurately. The effect of core-shell formation/suppression depending on Bi³⁺ stoichiometry has been reproduced by several synthesis runs in the last 4 years (some of the recently published works include: Acosta *et al.*, *J. Am. Ceram. Soc.* **e** 3405 (2015), Koruza *et al.*, *J. Eur. Ceram. Soc.* **36** 1009 (2016), Frömling *et al.* *J. Mater. Chem. C*, **6** 738 (2018)). We also corroborated the in situ formation of core-shell nanoparticles in several runs, two different heating stages, three different TEMs and more than one nanoparticle on each run. To our view, the experiments clearly show that in situ synthesis is not a requirement and reproducibility is not an issue. Nevertheless, it is worth to emphasize here that we have achieved to follow this core-shell formation process inside a transmission electron microscope for the first time.

Page 5: The authors wrote "The calculation shed lights on a new mechanism to minimize the domain size of ferroelectrics" and " Consequently, a small strain gradient can produce such star-like polar nanoregions. Producing a strain gradient near phase transitions in ferroelectrics can thus be used as a strategy to minimize domain size and concomitantly maximize electromechanical properties ". These statements are very misleading and far-reaching without any foundation.

We thank the reviewer for this comment. We may have included too far-reaching concepts without enough evidence. As the reviewer suggests, such statements are not in the scope of the manuscript. We decided to omit these claims and explore these concepts in future work.

Observation of small domains in this work (whose ferroelectric character has not been proven) at high temperatures in in-situ prepared samples does not imply that other ferroelectric nanoparticles with a strain gradient would exhibit small domains. Do authors know if in-situ prepared nanoparticles (used in this work) would be ferroelectric at room temperatures? Are they ferroelectric at 800°C? Would the same small domain-size be present at room temperature?

We performed preliminary experiments in which we investigate with piezo-force microscopy (PFM) the core-shell nanoparticles. We observe the difference in polarization between core and shell. There is presence of nanodomains at room temperature (predominantly in the core) and we did preliminary switching experiments. However, probing with certainty the ferroelectric character of these nanoparticles at room temperature (or 800 °C) is by no means trivial. One of the key challenges with the PFM experiments is that the nanoparticles geometry influences the measurements considerably and can lead to artifacts in the switching experiments. Thus, we decided to leave these experiments and insights for future contributions.

There is no general theory that shows that small domain size must lead to large response. That is a conjecture that has been often used in the literature (in some cases materials with a small domain size indeed show large response), but there

are opposite examples (Jin 2009, APL). I suggest to the authors to simply drop those claims as they are not central to their paper.

We agree with the reviewer on this point and decided to remove these claims, which are beyond the scope of this article.

In conclusion section, the authors wrote "Electric field and temperature dependent in situ TEM experiments demonstrated that the domain-like nanoregions are polar and that they can be switched."

Could authors explain better how experimental results demonstrated polar character of nano regions? Is it Figure 4 and how exactly?

As previously discussed, the observed splitting of the $(2\bar{2}0)$ reflex is the indicator for polarity.

Please refer to the response for the question raised in response letter page 4.

We would like to mention that the sentence referred by the reviewer is modified in the revised manuscript:

"Our results indicate that the polarization distribution is a direct consequence of nanoscale flexoelectricity and that they can be switched by an applied electric field."

Reviewer #2 (Remarks to the Author):

In their paper Dr. L. Molina-Luna et al., have claimed to realize switchable electrical polarization assisted by flexoelectricity in BNT-25ST core-shell nanoparticles. Flexoelectricity is a rapidly developing field, but so far, the center focus was exploring means of generating large nanoscale strain-gradient/polarization (non-switchable) and its applications. The successful demonstration of “switchable flexoelectric polarization”, therefore, will be an important advancement in this field. However, the experimental evidence and arguments presented in the current manuscript are not strong enough to support the authors’ claims.

1) According to their phase-field simulations, a strain-gradient of the order of 10^5 m^{-1} is necessary to stabilize domain-like polar nanoregions (in Fig. 3a). However, authors just provided a qualitative assessment of the strain-distribution based on the GPA analysis (Fig. S2). Thus, the authors could attempt to quantify the strain-gradient within the nanoparticles.

We thank the reviewer for this comment. We made an effort to quantify the local strain-gradient as suggested by the reviewer by using the data available from the image shown in Fig. 4a. This is not trivial since atomic resolution is needed to perform the task. We performed a noise-reduction treatment of the available HRTEM images and created region-of-interest boxes (ROI) of areas with visible atomic resolution information. Geometrical Phase Analysis (GPA) relies on measuring and mapping displacement fields and strain fields directly from HRTEM images (Hýtch, M. J. *et al.* Quantitative measurement of displacement and strain fields from HREM micrographs. *Ultramicroscopy* **74**, 131–146 (1998)).

By following the described procedure we generated geometrical phase analysis (GPA) evaluation maps and the corresponding ϵ_{xx} strain profiles of the core-shell interface area shown in Fig. S2. From the updated Fig.S2, it can be seen that the strain values within the nanoparticle change depending on the distance to the core area. The magnitude of the strain from the GPA and the phase field simulations are in the same order although the GPA analysis yields larger value because they are located at the interface while the phase field simulation provides the average effect, in other words, the effect of the core-shell is assumed to change linearly to simplify the calculations.

We also updated Fig. 4 with a local strain gradient along the core-shell interface as an inset.

2) Figures 4, S3, S5, and the supplementary video are most relevant in understanding the central claim made by the authors. I have following questions regarding these data.

a) Do I understand correctly, the “field-induced domain-like nanoregions” (line

127) are the same as those induced by the flexoelectric effect in Fig. 3a? If yes, why the reflex corresponding to the domain-like nanoregions only appears in the FFT image obtained under an electric field of 11 KV/mm but not in the image obtained under the zero electric field?

For the zero field, Fig. 3a and Fig. 4a show domain-like nanoregions which are induced by the flexoelectric effect. The reflexes observed correspond to lattice spacings in the nanoparticle.

For the case with applied field, the domain-like nanoregions shown in Fig. 4b and Fig. 4c are the result of both the flexoelectric effect and applied electric field. The applied electric field (e.g., 11 kV/mm) is capable of changing the crystal structure (field induced phase transformation) or promoting domain processes like switching or nucleation of new domain-like nanoregions. These processes alter the contrast in the TEM images and also the reflections in the FFT.

When the field is removed, the polarization is back to its original state and the corresponding FFT patterns vanish.

b) In line# 131 authors claimed, “Under the removal of the electric field, the domain-like nanoregions vanish.” Does it mean the polar nanoregions are unstable against electrical-cycling? If yes, why they are unstable?

We thank the reviewer for pointing out this misleading sentence. We changed the sentence to:

"The core and shell DLNRs are in a stable configuration at the given conditions."

(page 4)

The nanoregions are stable. There is a stable domain-like nanoregions configuration without application of electric field. What we meant is that the electric field-induced processes are reversible. We observe the reversibility even after several electrical cycles.

c) Based on the phase-field simulations, authors argued that for high field strength, the electric field-induced polarization overshadows the flexoelectric-polarization. This results in the mere reorientation of polarization inside the nanoparticle.

If only the polarization is reorienting, why the FFT images (in Fig. S3 and Supplementary video) corresponding to $E = -21.9$ KV/mm and $+21.9$ KV/mm look different? In the former case, alongside the primary reflexes, additional secondary reflexes can be observed. These secondary reflexes, however, are absent in the latter case. It will help if the authors clarify the origin and the relevance of this difference in the switching of polarization.

We thank the reviewer for pointing this out. We carefully re-checked all our data analysis. The Fig. S3 for -21.9 kV/mm was actually the FFT of the whole area (core and shell) in contrast to Fig. S3 for +21.9 kV/mm where we considered the regions separated. We now updated correspondingly the figures for consistency (both S3 and the supplementary video).

The FFTs were extracted from regions-of-interest (ROI) of exactly the same positions. As seen in the corrected images, the FFTs of $E = -21.9$ KV/mm and $+21.9$ KV/mm are similar. Minor discrepancies between positive and negative states under electric can be related to many variables that influence switching processes (local electric field, point defects, crystal structure, among others).

d) While discussing the effect of electrical biasing, authors do not consider the motion charged defects such as oxygen vacancies, which will be inevitably present in the nanoparticle given the inhomogeneous distribution of Sr^{2+} . At the operating temperature (800 C), the oxygen vacancies would be highly mobile, and electrical biasing would readily alter their spatial distribution. This could affect the strain-distribution and hence the flexoelectric effect inside the nanoparticle. Thus, authors should clarify the role of ionic motion in the electric field-induced polarization switching.

The reviewer raises an important and very interesting point; we agree that a gradient of oxygen vacancies could affect the strain-distribution. In our previous work (Frömling *et al. J. Mater. Chem. C*, **6** 738-744 (2018)) we actually measured impedance of bulk core-shell and non-core-shell materials at high-temperature. The resistivity of those bulk samples was still quite high even at 800 °C and we could not measure the standard semi-circles expected in Nyquist plots for oxygen conductors. Of course, this is only a macroscopic view.

Probing oxygen motion in nanoparticles at high-temperature is, to our knowledge, currently not possible. We thank the reviewer for the comment and we will keep this comment to try to address it in a future contribution.

We added the following sentence to the main manuscript to clarify this:

*"The role of oxygen vacancies under electric field should not be neglected. In recent work of Das *et al.*⁴⁰, controlled manipulation of oxygen vacancies in STO under mechanical loading was reported. In that case, flexoelectricity enabled the redistribution of oxygen vacancies. In our case, the oxygen vacancies may similarly influence the strain distribution and therefore the polarization by the flexoelectric effect. However, the resistivity of NBT-25BT bulk samples is relatively high even at 800 °C²³ and it was not possible to measure the standard semi-circles expected in Nyquist plots. In situ TEM measurements of such defects with atomic resolution at 800 °C would be even more demanding and falls out of the scope of the present article."*

(page 6)

e) Another important issue the authors have not discussed concerns the microscopic mechanism behind the electrical switching of polarization. In conventional ferroelectrics, the switchable polarization is due to the reversible ionic displacement under the applied electric field. In terms of the Landau energy landscape, this would correspond to switching between the two minima of the double well potential (Fig. S5). The authors showed in Fig. S5 that with increasing temperature the energy landscape changes from the double well into a single well. Therefore, it is not clear how the polarization switches in the absence of two potential minima?

We thank the reviewer for clarifying this point. We observe “domain switching” due to the applied field. The reason is the following:

The polarization in the ferroelectric material may come from different origins, e.g. spontaneous polarization in a ferroelectric state which can be described by the Landau polynomial, field-induced polarization (ϵE), strain induced polarization (piezoelectricity), strain-gradient induced polarization (flexoelectricity)

$$P \approx P^S + P^E + P^{piezo} + P^{flexo} = P^S + \epsilon_0 E + d\epsilon + f\nabla\epsilon$$

where ϵ_0 , d and f are dielectric piezoelectric and flexoelectric coefficients, respectively.

As the reviewer mentioned, in Fig. S5, we show that above the Curie temperature the Landau energy is single well. The corresponding spontaneous polarization P^S in this case is zero. At such temperature, piezoelectricity also vanishes, i.e., $P^{piezo} = 0$. The polarization here mainly consist of two parts, i.e. an electric field induced one and a strain gradient induced one.

$$P \approx P^E + P^{flexo}$$

As consequence of the radial strain distribution, the distribution of P^{flexo} has a complex configuration shown in Fig. 3e, where in some region the polarization point to the center while some others parts point to the outer. When applying the field, the polarization in the opposite direction reduces its magnitude. If the applied field is large enough, the polarization changes its direction.

However, the change of the polarization does not occur smoothly due to the gradient energy contribution. In the phase field model, H^{grad} in eq. (1) of manuscript represent the gradient energy. With this term considered, the polarization magnitude is influenced by the neighboring domains, which they tend to have the same magnitude and the same direction to minimize the total energy. If the field is opposite to the polarization, the change of the polarization is sharp (depending on the neighboring regions), as can be seen in Fig. 4e and 4f.

We mark the polarization change for the calculations corresponding to 10 and 20 kV/mm in Fig. R5. The regions pointed by the red arrow show the switching process of the nanodomains.

Figure R5. Comparison of polarization distribution with different applied field. The marked regions show the switching of the polarization.

Reviewer #3 (Remarks to the Author):

In this manuscript the authors reported domain-like nanopolar regions in NBT-25ST core shell nanoparticles at elevated temperature when ferroelectric polarization usually vanishes, and attribute this polarization to the flexoelectric effect that arise from the diffusion of Sr^{2+} cations. The authors used both in-situ TEM characterization method and phase-field simulation to prove their assumptions. However I found the following questions critical that impede my recommendation for its publication in nature communications.

(1) The authors observed Sr^{2+} cations diffusion from center to the edges of the nanoparticles in TEM, and interpret it to a linearly increasing isotropic eigenstrain in the model. To my knowledge this strain is called chemical Vegard strain and the effect is called Vegard's effect.

The Vegard effect gives rise to the eigenstrain associated with changes in the chemical stoichiometry. The eigenstrain used in the paper indeed originated from this Vegard effect. To acknowledge this, we included the reference: A. N. Morozovska *et al. Phy. Rev B* **83**, 195313 (2011) and A. N. Morozovska *et al. Phy. Rev B* **90**, 214103 (2014).

I think here the authors confuse Vegard chemical strain (which is due to the Vegard expansion of the lattice caused by mobile species) with the flexoelectric strain (which is induced from the polarization gradient $\epsilon_{ij} = F_{ijkl} * dP_k/dx_l$ via converse flexoelectric effect, see Eq. (13) in their own manuscript, also refer to paper *Phy. Rev. B* **83**, 195313 (2011)). Therefore based on their modeling, the ferroelectric polarization at elevated temperature is actually induced by Vegard's strain effect, NOT flexoelectric effect.

We argue that the Vegard strain is not the reason for the polarization. Instead, the strain gradient generated by the Vegard effect is the reason for the polarization. This is in fact the flexoelectric effect ($P_i = f_{ijkl} \frac{\partial \epsilon_{kl}}{\partial x_j}$). As a counterpart, homogeneous Vegard strain should not lead to polarization.

In addition to the eigenstrain due to the Vegard effect, we also included the flexoelectric strain due to the resultant polarization gradient in our simulation for completeness, but it is not responsible for the formation of the domain-like nanoregions.

We added the following text into the manuscript accordingly to avoid misunderstandings.

"A gradient of Sr^{2+} leads to a chemically induced lattice strain because of the differences in ionic radii of the A-site cations²³. This effect is usually referred as the Vegard effect^{32,33}."

"According to the Vegard law³³, the lattice parameter is linearly changed with the constituent's concentration. We treat the Vegard strain as the eigenstrain^{35,36} in the phase field simulation. The Sr^{2+} concentration is assumed

to increase linearly from the center to the edge. Hence, the eigenstrain is set to increase from the center to the edge accordingly, as defined in Eq. (6) and visualized in Fig. 3d. Figure 3e shows the calculated polarization induced by the strain gradient."

(page 3; page 3-4)

In order to clarify this point, we also provided the total strain distribution according to the eigenstrain we set, see Fig. R6 which appears as Fig. S7 of the supplementary material.

Figure R6. Mechanism of polarization generation in core-shell nanoparticles at elevated temperature starting from the chemical stoichiometry.

(2) In Figure S5, the authors studied different domain patterns by tuning the eigenstrain via coefficient W (Vegard expansion coefficients), while in Eq. (9) the author actually introduced flexoelectric energy H_{flexo} to the total free energy. If the domain-like pattern at 800 °C is induced from flexoelectric effect as they argued, the authors should turn on and off flexoelectric contribution by

setting $f_1=f_2=f_4=0\sim 10V$ to see the differences in domain pattern, rather than tuning the Vegard expansion coefficient (W). We believe this can clarify which effect is responsible for the observed behavior.

Figure S5 represents practically also the influence of different \mathbf{f} . In fact, increasing W (0, 1e4, 2e4) means increasing the strain gradient $\frac{\partial \varepsilon_{kl}}{\partial x_j}$ from 0 by a factor of 1 and 2, accordingly. Since the flexoelectric effect $P_i = f_{ijkl} \frac{\partial \varepsilon_{kl}}{\partial x_j}$, one can interpret the results in another way. More exactly, the results would be the same as in Fig. S5 if the flexocoupling coefficient \mathbf{f} increases from 0 by a factor of 1 and 2 respectively, for a fixed strain gradient.

Particularly, the results of the case without the flexoelectric effect ($\mathbf{f}=\mathbf{0}$) are given in Fig. S5, 1st row 3rd column. There are no nanodomains visible. We did the simulation with $\mathbf{f}=\mathbf{0}$, the results is exactly the same.

(3) The material of study in this paper is $0.75\text{Bi}_{1/2}\text{Na}_{1/2}\text{TiO}_3\text{-}0.25\text{SrTiO}_3$, However the Landau energy coefficients (alpha) used in the phase-field simulation are collected from the paper for BaTiO_3 , “Li, Y. L., Cross, L. E. & Chen, L. Q. A phenomenological thermodynamic potential for BaTiO_3 single crystals. *J. Appl. Phys.* 98, 64101 (2005).” Although the authors did modify the temperature dependent coefficients alpha1 based on different Curie temperature (813K instead of 115oC), the Curie constant and the other Landau energy coefficients are exactly the same. That probably explain why the calculated polarization in Fig. 3 and Fig. S5 are up to 0.26C/m², which is the reported spontaneous polarization of BaTiO_3 , rather than 0.38C/m² for BNT-ST. Therefore I doubt if the modeling results based on the BTO coefficients can explain the real experimental observations in BNT-ST.

We thank the reviewer for this comment. We adopted in the revision three different sets of Landau energy coefficients (for BTO, PTO, NBT-25ST; Fig. R9-11). Results show that the parameters have little influence on the pattern of the flexoelectric-induced polarization distribution above the Curie temperature, while below Curie temperature the domain structures vary considerably.

The parameters for BTO and PTO are from Chen, L.-Q. APPENDIX A – Landau Free-Energy Coefficients. *Physics of Ferroelectrics* 363–372. The Landau parameters for NBT-25ST are derived in the following.

First, we assume the Landau energy under mechanical stress-free boundary condition taking the form of a sixth-order polynomial:

$$H^{bulk} = \alpha_1(P_1^2 + P_2^2) + \alpha_{11}(P_1^4 + P_2^4) + \alpha_{12}(P_1^2 P_2^2) + \alpha_{111}(P_1^6 + P_2^6) + \alpha_{112}(P_1^4 P_2^2 + P_2^4 P_1^2)$$

The coefficient α_1 is temperature dependent coefficient. For ferroelectrics, it is related to the permittivity by the Curie-Weiss law. In the paraelectric phase:

$$2\alpha_1 = \frac{\partial^2 H^{bulk}}{\partial P_1^2} = \frac{\partial E_1}{\partial P_1} = \frac{1}{\varepsilon} = \frac{T - T_0}{C} = 2\alpha_0(T - T_0)$$

where T_0 is the Curie-Weiss temperature and C is the Curie constant.

However, for relaxors, (e.g. NBT-25ST), the Curie-Weiss law is no longer valid. Instead, the permittivity follows:

$$\frac{1}{\varepsilon} - \frac{1}{\varepsilon_m} = \frac{(T - T_m)^\gamma}{C}$$

where γ is the diffusion factor which takes the value between 1 and 2. (Viehland, Dwight, and Jie Fang Li. *Journal of applied physics* **74**(6), 4121-4124 (1993)) We used the permittivity-temperature graph from Gomah-Pettry *et al.* (Gomah-Pettry Jean-Richard, *et al. Journal of the European Ceramic Society* **24**(6), 1165-1169 (2004)) and fit it to the above modified Curie-Weiss law, the results are shown in Fig. R7. The diffusion factor is $\gamma = 1.004$, which means that the dielectric behavior (at high frequency) of this material is more like a ferroelectric. As a simplification, we assume $\gamma = 1$. The Landau coefficient is:

$$\alpha_0 = 5.14 \times 10^5 (T - 216)$$

Then we attempt to determine the other coefficients.

At Curie temperature, the Landau energy for paraelectric phase and ferroelectric phase are the same, i.e.:

$$\alpha_0(T_c - T_0)P_{sc}^2 + \alpha_{11}P_{sc}^4 + \alpha_{111}P_{sc}^6 = 0$$

where P_{sc} is the spontaneous polarization at T_c .

On the other hand, at T_c :

$$\left. \frac{\partial H^{bulk}}{\partial P} \right|_{P=P_{sc}} = 2\alpha_0(T_c - T_0)P_{sc} + 4\alpha_{11}P_{sc}^3 + 6\alpha_{111}P_{sc}^5 = 0$$

Combining the above two equations, one obtains:

$$T_c - T_0 = \frac{-\alpha_{11}^2}{2\alpha_{111}\alpha_0}$$

$$P_{sc}^2 = \frac{-\alpha_{11}}{2\alpha_{111}}$$

The value of P_{sc} is obtained in (Krauss Werner, *et al. Journal of the European Ceramic Society* **30**, 1827-1832 (2010)). We regard the temperature with maximum permittivity T_m as T_c . Put all these values to the above equations, we get $\alpha_{11} = -1.25 \times 10^{10} C^{-4} m^6 N$ and $\alpha_{111} = 3.91 \times 10^{12} C^{-6} m^{10} N$.

The value of α_{12} and α_{112} depends on the transition temperature between orthorhombic and tetragonal phases. For NBT-25ST, there is no such phase

transition. According Watanabe *et al.*, the rhombohedral-tetragonal phase transition temperature for NBT-25ST is 40 °C. If one assumes that 40 °C is the orthorhombic-tetragonal phase transition temperature, then:

$$G(P_1 = 0; P_2 = P_t) = G(P_1 = P_t; P_2 = 0) = G(P_1 = P_2 = P_o)$$

and

$$\left. \frac{\partial H^{bulk}}{\partial P_1} \right|_{P_1=P_2=P_o} = 2\alpha_0(T_c - T_0)P_0 + 4\alpha_{11}P_0^3 + 6\alpha_{111}P_0^5 + 2\alpha_{12}P_0^3 + 6\alpha_{112}P_0^5 = 0$$

where P_t and P_o are the components of the spontaneous polarization in one direction for the tetragonal and the orthorhombic phase, respectively. By solving these two relationships simultaneously, one obtains the value of $\alpha_{12} = 3.52 \times 10^{10} C^{-4} m^6 N$ and $\alpha_{112} = -1.25 \times 10^{12} C^{-6} m^{10} N$. Fig. R9 shows the Landau energy as a function of P_1 and P_2 . The color indicates the Landau free energy (H^{bulk}). Three temperatures were chosen, 100 °C (at the tetragonal phase) 200 °C (near Curie temperature) and 800 °C (the probing temperature of the experiments) for the phase field simulations. The results of the simulation are shown in Fig. R10. The polarization configuration is almost the same to the previous calculation based on the modified BTO parameter, despite the magnitude of the polarization in the tetragonal phase. The three simulation results demonstrate that our simulation and conclusion points are still valid at high-temperature despite the different Landau parameters chosen.

To make our simulation more comparable to the observation, we modified Fig. 3 of the main manuscript and Fig. S5 accordingly with the updated NBT-25ST parameters.

Figure R7. Fitting the susceptibility as a function of temperature in logarithmic coordinates by using the modified Curie-Weiss law for NBT-25ST. Blue dot: experimental data from Jean-Richard G., *et al. J. Eur. Ceram. Soc.* **24**(6), 1165-1169 (2004)); red line: fitting curve.

Figure R8. The Landau energy as a function of the polarization by using the derived coefficients of NBT-25ST. The rhombohedral-tetragonal-cubic phase transition can be captured.

Figure R9. Polarization distribution of a nanoparticle with NBT-25ST Landau coefficient at different temperatures and eigenstrain values.

Figure R10. Polarization distribution of a nanoparticle with BTO Landau coefficient.

Figure R11. Polarization distribution of a nanoparticle with PTO Landau coefficient.

(4) The authors argued that the center of the nanoparticle is under compressive strain and the edge under tensile strain when the Sr diffuse from the center to the edges, which is counter-intuitive to me. Shouldn't more Sr ions at the edges induce compressive strain and the loss of Sr ions at the center induce tensile strain?

We thank the reviewer for pointing out this confusion of different strain contributions. In fact, the total strain is superposition of two contributions. One is the eigenstrain which correlates with the Sr^{2+} concentration and increases from the core to the shell. The other is the elastic strain, which results from the non-uniform eigenstrain and is positive (tensile) in core and negative (compressive) in shell. Nevertheless, the total strain in the shell can still be positive, if there the positive eigenstrain overtakes the negative elastic strain.

The related sentence in the original manuscript is changed to

"In this case, the magnitude of the eigenstrain increases from the center to the edge."

(page 3)

(5) In Figure 3(c) and Figure S5 (lower-right one), the scale bars do not match each other.

We have created an updated Fig. 3 and Fig. S5 accordingly. Thank you for pointing this out.

Based on these concerns, I cannot recommend it for publication in nature communications at its current form.

We are very grateful to the reviewer for all his/her insightful comments and hope we have improved the manuscript in such a way that it can be recommended for publication.

Reviewers' comments:

Reviewer #3 (Remarks to the Author):

The authors now clarify the inhomogeneous ionic distribution causes the local strain via the Vegard effect, which later translates into the polarization via the flexoelectric effect. The authors now clearly point that out in the revised manuscript. This also explains my previous concerns that changing the Vegard strain coefficients (W) in the phase-field model has the same effect as changing the magnitude of flexoelectric effect (Figure S5).

I am also impressed by the author's effort to try to fit the Landau energy coefficients of NBT-ST in phase-field model, and compare the effect of using different coefficients (BTO, PTO, NBT-ST). Since the phase-field model is done on a 2D geometry (where rhombohedral and orthorhombic phases cannot be well represented in the 2D model, by tetragonal phase is fine), I am thinking maybe 4th order polynomial is sufficient for this work. Chen L.Q. et. al. recently published in APL on how to fit Landau energy coefficients for ferroelectrics which the authors might find useful.

The 2nd reviewer's 1st question is actually a very good one. If the strain distribution can be quantitatively evaluated in TEM, then the flexoelectric effect can be directly measured. I believe this information will be very useful and can be directly fed into the phase-field model to calculate the flexoelectric induced polarization (right now the model is based on a rough assumption that Sr composition and thus the Vegard eigenstrain is linear increasing from core to shell). I agree with the authors' response that quantification of strain gradient is nontrivial. However I would bring to the author's attention that direct measurement of flexoelectric polarization at the atomic scale was indeed done by P. Gao's group and the results were published on PRL recently. (<https://journals.aps.org/prl/abstract/10.1103/PhysRevLett.120.267601>)

Also I agree with one of the first reviewer comments. The argument that mechanical tip pressure is not necessary to induce ferroelectric polarization via flexoelectric effect has been claimed before and should not be considered as the novelty in this manuscript. Lane Martin's group has published quite a lot on compositional graded ferroelectrics. Catalan and Noheda's Nature Material paper and Gao P.'s PRL are other examples.

Overall, I am satisfied with the authors' response and revision on the manuscript. I would recommend it for publication in nature communications.

Reviewer #4 (Remarks to the Author):

I have carefully read the revised manuscript by L. Molina-Luna et al. and their response to the reviewers' comments. This paper reports on designing polar materials by flexoelectricity that display switchable polarization at extreme temperature. This is an interesting and important discovery that might lead to new technological avenues and would be in the interest of a broader community. In addition, I think most of the reviewers' comments have been addressed constructively. But, there're still some issues that authors should clarify before publication.

1) In line #117 of the manuscript, the authors wrote: " We observed DLNRs within the core and shell regions, which are magnified to aid visualization in the top insets" . It is very hard to see DLNRS within the magnified view of the shell region (Fig. 4a top inset). On the contrary, some pattern is discernible in the magnified view of the core region.

2) I would also suggest using scale bars in the top insets of Fig. 4.

3) Authors attributed the splitting of (2-20) reflex under the electric field to two prevalent

polarities (Fig. 4c and Fig. S3). Authors should properly discuss what they mean by these two polarities. From the phase field simulation provided by the authors, one might argue to see a quasi-uniform polarization distribution under high electric field (Fig. 4f).

4) Furthermore, this splitting is very confusing. If we look at Fig. 4c, Fig. S3, and the supplementary video together, the splitting sometimes appears in the FFT image sometimes does not. For example, the splitting is evident in Fig. 4c and Fig. S3 for $E = +21.9$ KV/mm. However, this splitting is absent in the supplementary video for $E = +21.9$ KV/mm. Authors must clarify this confusion; it goes against their response to one of the reviewer's concerns, where they claimed that the electric-field-induced processes are reversible. However, the difference is obvious.

5) I am bit surprised by the authors' choice of the flexocoupling coefficients, which generally are in the range of 1-10 V for titanates. Also, I could not find the values used by the authors in the paper they referred (Ref. 43, Ponomareva, I. et al. PRB (2012)). I must say, though, the mistake could be entirely in my part.

Reviewers' comments:

Reviewer #3 (Remarks to the Author):

The authors now clarify the inhomogeneous ionic distribution causes the local strain via the Vegard effect, which later translates into the polarization via the flexoelectric effect. The authors now clearly point that out in the revised manuscript. This also explains my previous concerns that changing the Vegard strain coefficients (W) in the phase-field model has the same effect as changing the magnitude of flexoelectric effect (Figure S5).

I am also impressed by the author's effort to try to fit the Landau energy coefficients of NBT-ST in phase-field model, and compare the effect of using different coefficients (BTO, PTO, NBT-ST). Since the phase-field model is done on a 2D geometry (where rhombohedral and orthorhombic phases cannot be well represented in the 2D model, by tetragonal phase is fine), I am thinking maybe 4th order polynomial is sufficient for this work. Chen L.Q. et. al. recently published in APL on how to fit Landau energy coefficients for ferroelectrics which the authors might find useful.

We thank the reviewer for his/her recognition of the improved manuscript.

We thank the reviewer for helping us track the most recent work in this field. The method used for the determination of the Landau parameter for BST can be implemented for other systems. We plan to refer to this in future contributions.

We also believe that the 4th order polynomial (if it is a double well shape below T_c and single well at high temperature) is sufficient for this work. However, we would like to mention here that, as one can see from our revised version, we made an effort to directly provide the Landau parameter for BNT-25ST using the experimental data (including transition temperature, spontaneous polarization etc.). The identification of the Landau parameters for this particular system has never been provided before. Thus, we believe that by providing this other researchers in this field can benefit.

The 2nd reviewer's 1st question is actually a very good one. If the strain distribution can be quantitatively evaluated in TEM, then the flexoelectric effect can be directly measured. I believe this information will be very useful and can be directly fed into the phase-field model to calculate the flexoelectric induced polarization (right now the model is based on a rough assumption that Sr composition and thus the Vegard eigenstrain is linear increasing from core to shell). I agree with the authors' response that quantification of strain gradient is nontrivial. However I would bring to the author's attention that direct measurement of flexoelectric polarization at the atomic scale was indeed done by P. Gao's group and the results were published on PRL recently.

<https://journals.aps.org/prl/abstract/10.1103/PhysRevLett.120.267601>

We thank the reviewer for providing this very recently published paper. The direct measurement of the polarization induced by dislocations via the flexoelectric effect at the atomic scale is indeed highly related to this work. Thus, we would like include this into our manuscript and rephrase the related sentences in page 3:

“As shown in recent work on strontium titanate, atomic scale measurements of local displacements due to the flexoelectric effect have been reported³⁴. However, for the BNT-25ST nanoparticle system, the measurement of atomic-displacements for the whole nanoparticle is nontrivial. Nevertheless, a quantitative assessment of the total strain distribution by atomic-displacement mapping in small regions of interest is possible (see Supplementary Fig. S2), which indicates large strain gradients within the single nanoparticle. The value of strain ranges from -0.2% to 0.2 % at the distance of 3.8 nm.”

Also I agree with one of the first reviewer comments. The argument that mechanical tip pressure is not necessary to induce ferroelectric polarization via flexoelectric effect has been claimed before and should not be considered as the novelty in this manuscript. Lane Martin's group has published quite a lot on compositional graded ferroelectrics. Catalan and Noheda's Nature Material paper and Gao P.'s PRL are other examples.

We thank the reviewer for this comment. As one can see from our revised manuscript (more specifically the second paragraph), we have now introduced more accurately the novelty of the work and discussed carefully the literature on the topic.

Overall, I am satisfied with the authors' response and revision on the manuscript. I would recommend it for publication in nature communications.

We thank again the reviewer for his/her contribution to the paper.

Reviewer #4 (Remarks to the Author):

I have carefully read the revised manuscript by L. Molina-Luna et al. and their response to the reviewers' comments. This paper reports on designing polar materials by flexoelectricity that display switchable polarization at extreme temperature. This is an interesting and important discovery that might lead to new technological avenues and would be in the interest of a broader community. In addition, I think most of the reviewers' comments have been addressed constructively. But, there're still some issues that authors should clarify before publication.

1) In line #117 of the manuscript, the authors wrote: "We observed DLNRs within the core and shell regions, which are magnified to aid visualization in the top insets". It is very hard to see DLNRs within the magnified view of the shell region (Fig. 4a top inset). On the contrary, some pattern is discernible in the magnified view of the core region.

For Fig. 4a, DLNRs are purely induced by the strain-gradient within the nanoparticle. For the core region, especially near the core-shell interface, the lattice is highly strained, thus DLNRs can be easily captured. While in the shell region we selected (red box), the DLNRs are not obvious. In other parts, for instance, bottom-left region of the particle, DLNRs are more obvious. We specifically choose this red box region, in order to compare the two states: with and without the electric field. (Compare Fig 4a to 4b-c). To make our sentences more rigorous, we limited our phrase in the following way in Page 4:

"Though not obvious in some regions in the shell, we observed DLNRs within the nanoparticle, in which the selected regions are magnified to aid visualization in the top insets."

2) I would also suggest using scale bars in the top insets of Fig. 4.

We thank the reviewer for this remark and added a scale bar to the inset in the first row.

3) Authors attributed the splitting of (2-20) reflex under the electric field to two prevalent polarities (Fig. 4c and Fig. S3). Authors should properly discuss what they mean by these two polarities. From the phase field simulation provided by the authors, one might argue to see a quasi-uniform polarization distribution under high electric field(Fig. 4f).

We thank the reviewer for pointing this out. The splitting of a single point in the FFT correlates to the different distorted crystal structures due to different strain states. To be specific, the FFT pattern marked by the white arrow in Fig. 4c (where one can find two splitting points) indicates the existence of two polarities throughout the region of interest (ROI), here the top insets. Each of the split points provides the information of either dark or bright region shown in the top inset.

For the second question, the TEM image and the corresponding FFT provides detailed structural information about the nanoparticle, including the atomic displacement and longer-range order. The phase field simulation is able to provide the average polarization in the simulated element, but it is not able to reflect the change of each atomic position, thus missing the splitting of the different polarity states.

To clarify this for the audience, we modified the related sentence in Page 4:

“This indicates that there are two different polarities prevalent, e.g. the dark and bright regions as shown in the top insets.”

4) Furthermore, this splitting is very confusing. If we look at Fig. 4c, Fig. S3, and the supplementary video together, the splitting sometimes appears in the FFT image sometimes does not. For example, the splitting is evident in Fig. 4c and Fig. S3 for $E = +21.9$ KV/mm. However, this splitting is absent in the supplementary video for $E = +21.9$ KV/mm. Authors must clarify this confusion; it goes against their response to one of the reviewer’s concerns, where they claimed that the electric-field-induced processes are reversible. However, the difference is obvious.

We thank the reviewer for his careful check of the video.

Figures 4c and S3 provide the FFT of the same position shown in the red and blue box.

We rechecked the last frame of the video and used the FFT for the exactly the same region for the video. We hope the updated video is able to give a better and correct overview of the in situ experiment.

Here, we also want to emphasize two points:

- 1) The reversibility of our process does not necessarily mean that the FFT for -21.9 kV/mm and $+21.9$ kV/mm should be exactly the same. It mainly indicates that by applying an electric field (no matter in which direction), the FFT of Fig. 4a (or the middle image Fig. S3) changes, and is recovered when the electric field is removed. This phenomenon is obvious from the experiments.
- 2) For a comparatively large electric field, the splitting shown in the FFT may vanish due to the dominance of the field-induced polarization (e.g. for -21.9 kV/mm case in the video). Nevertheless, for the next frame (-15.3 kV/mm), the splitting appears. This also supports the reversibility of the process.

5) I am bit surprised by the authors’ choice of the flexocoupling coefficients, which generally are in the range of 1-10 V for titanates. Also, I could not find the values used by the authors in the paper they referred (Ref. 43, Ponomareva, I. et al. PRB (2012)). I must say, though, the mistake could be entirely in my part.

We thank the reviewer for the careful check of the parameters used for the simulations.

The flexocoupling coefficients for titanate-based ferroelectric material varies from experimental values to first-principle calculations, especially for BTO (compare Ma and Cross *A.P.L.* (2016) with Maranganti and Sharma *P.R.B.* (2009)). Here, we decided to use the parameters for STO, where the experimental measurement and *ab initio* results are close to each other (See Zubko *et al.* *P.R.L.* (2007) and Maranganti and Sharma *P.R.B.* (2009)). The *ab initio* calculations provides values for the flexoelectric constants:

$$\mu_{11} = -26.4, \mu_{12} = -374.7 \mu_{44} = -357.9 (10^{-13} \text{ C/cm}),$$

and the experimental data yields:

$$\mu_{11} = 20, \mu_{12} = 700 \mu_{44} = 300 (10^{-13} \text{ C/cm}).$$

The flexoelectric constants can be transformed into the flexocoupling coefficients by the following equation:

$$f = \mu / (\epsilon_r \epsilon_0)$$

With the provided experimental data, we obtained our flexocoupling coefficients: $f_{11} = 0.0207$, $f_{12} = 0.721$, $f_{44} = 0.308$ (V). These coefficients are also adopted by Chen *et al. Acta Mech.* (2014).

Compared with the parameter for BTO, these sets of parameters are smaller. Even with such small flexocoupling coefficients, the flexoelectric-induced polarization cannot be ignored, and thus supports our argument in the manuscript.

Regarding the second question, in Ref. 43 (Ponomareva, I. *et al. P.R.B.* (2012)), Ponomareva *et al.* compared their results with the ones obtained by Zubko *et al.* and found that they are in good agreement. Thanks to the reviewer's careful check, we realized that it is not appropriate to cite this reference here. Thus, we replace it with the reference of Zubko and Chen, and we modified the "methods" part accordingly in page 18:

*"The flexocoupling coefficients have three non-trivial independent components, f_{11} , f_{12} and f_{44} , where $f_{11} = f_{1111} = f_{2222}$, $f_{12} = f_{1122} = f_{2211}$ and $f_{44} = f_{1212} = f_{2121}$. The flexocoupling coefficients f_{11} , f_{12} and f_{44} were set to 0.02 V, 0.7 V and 0.3 V, respectively according to the work on strontium titanate by Zubko *et al.*⁴⁴ and Chen *et al.*⁴⁵"*

We are very grateful to all the reviewers for their insightful comments. We hope that the second revision of the manuscript can be recommended for publication.

REVIEWERS' COMMENTS:

Reviewer #4 (Remarks to the Author):

I thank the authors for their response to my comments. I recommend the paper for publication in Nature Communications.